# ADVERSARIAL INFORMATION FACTORIZATION

## ABSTRACT

We propose a novel generative model architecture designed to learn representations for images that factor out a single attribute from the rest of the representation. A single object may have many attributes which when altered do not change the identity of the object itself. Consider the human face; the identity of a particular person is independent of whether or not they happen to be wearing glasses. The *attribute* of wearing glasses can be changed without changing the *identity* of the person. However, the ability to manipulate and alter image attributes without altering the object identity is not a trivial task. Here, we are interested in learning a representation of the image that separates the identity of an object (such as a human face) from an attribute (such as 'wearing glasses'). We demonstrate the success of our factorization approach by using the learned representation to synthesize the same face with and without a chosen attribute. We refer to this specific synthesis process as image attribute manipulation. We further demonstrate that our model achieves competitive scores, with state of the art, on a facial attribute classification task.

## 1 INTRODUCTION

Latent space generative models, such as generative adversarial networks (GANs) (Goodfellow et al., 2014; Radford et al., 2015) and variational autoencoders (VAEs) (Rezende et al., 2014; Kingma & Welling, 2013), learn a mapping from a latent encoding space to a data space, for example, the space of natural images. It has been shown that the latent space learned by these models is often organized in a near-linear fashion (Radford et al., 2015; Kingma & Welling, 2013), whereby neighbouring points in latent space map to similar images in data space. Certain "directions" in latent space correspond to changes in the intensity of certain attributes. In the context of faces, for example, directions in latent space would correspond to the extent to which someone is smiling. This may be useful for image synthesis where one can use the latent space to develop new design concepts (Dosovitskiy et al., 2017; Zhu et al., 2016), edit an existing image (Zhu et al., 2016) or synthesize avatars (Wolf et al., 2017; Taigman et al., 2016). This is because semantically meaningful changes may be made to images by manipulating the latent space (Radford et al., 2015; Zhu et al., 2016; Larsen et al., 2016).

One avenue of research for latent space generative models has been class conditional image synthesis (Chen et al., 2016; Odena et al., 2016; Mirza & Osindero, 2014), where an image of a particular object category is synthesized. Often, object categories may be sub-divided into fine-grain sub-categories. For example, the category "dog" may be split into further sub-categories of different dog breeds. Work by Bao et al. (2017) propose latent space generative models for synthesizing images from fine-grained categories, in particular for synthesizing different celebrities' faces conditional on the identity of the celebrity.

Rather than considering fine-grain categories, we propose to take steps towards solving the different, but related problem of image attribute manipulation. To solve this problem we want to be able to synthesize images and only change one element or attribute of its content. For example, if we are synthesizing faces we would like to edit whether or not a person is smiling. This is a different problem to fine-grain synthesis; we want to be able to synthesize two faces that are similar, with only a single chosen attribute changed, rather than synthesizing two different faces. The need to synthesis two faces that are similar makes the problem of image attribute manipulation more difficult than the fine-grain image synthesis problem; we need to learn a latent space representation that separates an object category from its attributes.

In this paper, we propose a new model that learns a factored representation for faces, separating attribute information from the rest of the facial representation. We apply our model to the CelebA (Liu et al., 2015) dataset of faces and control several facial attributes.

Our contributions are as follows:

1. Our **core contribution** is the novel cost function for training a VAE encoder to learn a latent representation which factorizes binary facial attribute information from a continuous identity representation (Section 3.2).

2. We provide an **extensive quantitative** analysis of the contributions of each of the many loss components in our model (Section 4.2).

3. We obtain classification scores that are competitive with state of the art (Zhuang et al., 2018) using the classifier that is already incorporated into the encoder of the VAE (Section 4.3).

4. We provide qualitative results demonstrating that our latent variable, generative model may be used to successfully edit the 'Smiling' attribute in more than $90\%$ of the test cases (Section 4.4).

5. We discuss and clarify the distinction between conditional image synthesis and image attribute editing (Section 5).

6. We present code to reproduce experiments shown in this paper: (provided after review).

## 2 LATENT SPACE GENERATIVE MODELS

Latent space generative models come in various forms. Two state-of-art generative models are Variational Autoencoders (VAE) (Rezende et al., 2014; Kingma & Welling, 2013) and Generative Adversarial Networks (GAN). Both models allow synthesis of novel data samples from latent encodings, and are explained below in more detail.

### 2.1 VARIATIONAL AUTOENCODER (VAE)

Variational autoencoders (Kingma & Welling, 2013; Rezende et al., 2014) consist of an encoder $q_\phi(z|x)$ and decoder $p_\theta(x|z)$; oftentimes these can be instantiated as neural networks, $E_\phi(\cdot)$ and $D_\theta(\cdot)$ respectively, with learnable parameters, $\phi$ and $\theta$. A VAE is trained to maximize the evidence lower bound (ELBO) on $\log p(x)$, where $p(x)$ is the data-generating distribution. The ELBO is given by:

$$\log p(x) \geq \mathbb{E}_{q_\phi(z|x)} \log p_\theta(x|z) - KL[q_\phi(z|x)||p(z)] \tag{1}$$

where $p(z)$ is a chosen prior distribution such as $p(z) = \mathcal{N}(\mathbf{0}, I)$. The encoder predicts, $\mu_\phi(x)$ and $\sigma_\phi(x)$ for a given input $x$ and a latent sample, $\hat{z}$, is drawn from $q_\phi(z|x)$ as follows: $\epsilon \sim \mathcal{N}(\mathbf{0}, I)$ then $\hat{z} = \mu_\phi(x) + \sigma_\phi(x) \odot \epsilon$. By choosing a multivariate Gaussian prior, the $KL$-divergence may be calculated analytically (Kingma & Welling, 2013). The first term in the loss function is typically approximated by calculating the reconstruction error between many samples of $x$ and $\hat{x} = D_\theta(E_\phi(x))$.

New data samples, which are not present in the training data, are synthesised by first drawing latent samples from the prior, $z \sim p(z)$, and then drawing data samples from $p_\theta(x|z)$. This is equivalent to passing the $z$ samples through the decoder, $D_\theta(z)$.

VAEs offer both a generative model, $p_\theta(x|z)$, and an encoding model, $q_\phi(z|x)$, which are useful as starting points for image editing in the latent space. However, samples drawn from a VAE are often blurred (Radford et al., 2015).

### 2.2 GENERATIVE ADVERSARIAL NETWORKS (GAN)

An alternative generative model, which may be used to synthesize much sharper images, is the Generative Adversarial Network (GAN) (Goodfellow et al., 2014; Radford et al., 2015). GANs consist of two models, a generator, $G_\theta(\cdot)$, and a discriminator, $C_\chi(\cdot)$, both of which may be implemented using

convolutional neural networks (Radford et al., 2015; Denton et al., 2015). GAN training involves these two networks engaging in a mini-max game. The discriminator, $C_\chi$, is trained to classify samples from the generator, $G_\theta$, as being 'fake' and to classify samples from the data-generating distribution, $p(x)$, as being 'real'. The generator is trained to synthesize samples that confuse the discriminator; that is, to synthesize samples that the discriminator cannot distinguish from the 'real' samples. The objective function is given by:

$$\min_\chi \max_\theta \mathbb{E}_{p(x)}[\log(C_\chi(x)] + \mathbb{E}_{p_g(x)}[\log(1 - C_\chi(x))] \tag{2}$$

where $p_g(x)$ is the distribution of synthesized samples, sampled by: $z \sim p(z)$, then $x = G_\theta(z)$, where $p(z)$ is a chosen prior distribution such as a multivariate Gaussian.

## 2.3 BEST OF BOTH GAN AND VAE

The vanilla GAN model does not provide a simple way to map data samples to latent space. Although there are several variants on the GAN that do involve learning an encoder type model (Dumoulin et al., 2016; Donahue et al., 2016; Li et al., 2017), only the approach presented by Li et al. (2017) allows data samples to be faithfully reconstructed. The approach presented by Li et al. (2017) requires adversarial training to be applied to several high dimensional distributions. Training adversarial networks on high dimensional data samples remains challenging (Arjovsky & Bottou, 2017) despite several proposed improvements (Salimans et al., 2016; Arjovsky et al., 2017). For this reason, rather than adding a decoder to a GAN, we consider an alternative latent generative model that combines a VAE with a GAN. In this arrangement, the VAE may be used to learn an encoding and decoding process, and a discriminator may be placed after the decoder to ensure higher quality of the data samples outputted from the decoder. Indeed, there have been several suggestions on how to combine VAEs and GANs (Bao et al., 2017; Larsen et al., 2016; Mescheder et al., 2017) each with a different structure and set of loss functions, however, none are designed specifically for attribute editing.

The content of image samples synthesized from a vanilla VAE or GAN depends on the latent variable $z$, which is drawn from a specified random distribution, $p(z)$. For a well-trained model, synthesised samples will resemble samples in the training data. If the training data consists of images from multiple categories, synthesized samples may come from any, or possibly a combination, of those categories. For a vanilla VAE, it is not possible to choose to synthesize samples from a particular category. However, conditional VAEs (and GANs) (Chen et al., 2016; Odena et al., 2016; Mirza & Osindero, 2014) provide a solution to this problem as they allow synthesis of class-specific data samples.

## 2.4 CONDITIONAL VAES

Autoencoders may be augmented in many different ways to achieve category-conditional image synthesis (Bao et al., 2017). It is common to append a one-hot label vector, $y$, to inputs of the encoder and decoder (Sohn et al., 2015). However, for small label vectors, relative to the size of the inputs to the encoder and the decoder model, it is possible for the label information, $y$, to be ignored.
[1]. A more interesting approach, for conditional (non-variational and semi-supervised) autoencoders is presented by Makhzani et al. (2015), where the encoder outputs both a latent vector, $\hat{z}$, and an attribute vector, $\hat{y}$. The encoder is updated to minimize a classification loss between the true label, $y$, and $\hat{y}$. We incorporate a similar architecture into our model with additional modifications to the training of the encoder for the reasons explained below.

There is a drawback to incorporating attribute information in the way described above (Makhzani et al., 2015; Perarnau et al., 2016) when the purpose of the model is to edit specific attributes, rather than to synthesize samples from a particular category. We observe that in this *naive* implementation of conditional VAEs, varying the attribute (or label) vector, $\hat{y}$, for a fixed $\hat{z}$ can result in unpredictable changes in synthesized data samples, $\hat{x}$. Consider for example the case where, for a fixed $\hat{z}$, modifying $\hat{y}$ does not result in any change in the intended corresponding attribute. This suggests that

---

[1]The label information in $y$ is less likely to be ignored when $y$ has relatively high dimensions compared to $z$ (Yan et al., 2016; Perarnau et al., 2016).

information about the attribute one wishes to edit, $y$, is partially contained in $\hat{z}$ rather than solely in $\hat{y}$. Similar problems have been discussed and addressed to some extent in the GAN literature (Chen et al., 2016; Mirza & Osindero, 2014; Odena et al., 2016), where it has been observed that label information in $\hat{y}$ is often ignored during sample synthesis.

In general, one may think that $\hat{z}$ and $\hat{y}$ should be independent. However, if attributes, $y$, that should be described by $\hat{y}$ remain unchanged for a reconstruction where only $\hat{y}$ is changed, this suggests that $\hat{z}$ contains most of the information that should have been encoded within $\hat{y}$. We propose a process to separate the information about $y$ from $\hat{z}$ using a mini-max optimization involving $y$, $\hat{z}$, the encoder $E_\phi$, and an auxiliary network $A_\psi$. We refer to our proposed process as 'Adversarial Information Factorization'.

## 2.5 ADVERSARIAL INFORMATION FACTORIZATION

For a given image of a face, $x$, we would like to describe the face using a latent vector, $\hat{z}$, that captures the identity of the person, along with a single unit vector, $\hat{y} \in [0, 1]$, that captures the presence, or absence, of a single desired attribute, $y$. If a latent encoding, $\hat{z}$, contains information about the desired attribute, $y$, that should instead be encoded within the attribute vector, $\hat{y}$, then a classifier should be able to accurately predict $y$ from $\hat{z}$. Ideally, $\hat{z}$ contains no information about $y$ and so, ideally, a classifier should not be able to predict $y$ from $\hat{z}$. We propose to train an auxiliary network to predict $y$ from $\hat{z}$ accurately while updating the encoder of the VAE to output $\hat{z}$ values that cause the auxiliary network to fail. If $\hat{z}$ contains no information about the desired attribute, $y$, that we wish to edit, then the information can instead be conveyed in $\hat{y}$ since $\hat{x}$ must still contain that information in order to minimize reconstruction loss. We now formalize these ideas.

## 3 METHOD

In what follows, we explain our novel approach to training the encoder of a VAE, to factor (separate) out information about $y$ from $\hat{z}$, such that $H(y|\hat{z}) \approx H(y)$. We integrate this novel factorisation method into a VAE-GAN. The GAN component of the model is incorporated only to improve image quality. Our main contribution is our proposed adversarial method for factorising the label information, $y$, out of the latent encoding, $\hat{z}$.

### 3.1 MODEL ARCHITECTURE

A schematic of our architecture is presented in Figure 1. In addition to the encoder, $E_\phi$, decoder, $D_\theta$, and discriminator, $C_\chi$, we introduce an auxiliary network, $A_\psi : \hat{z} \to \tilde{y}$, whose purpose is described in detail in Section 3.2. Additionally, the encoder also acts as a classifier, outputting an attribute vector, $\hat{y}$, along with a latent vector, $\hat{z}$.

The parameters of the decoder, $\theta$, are updated with gradients from the following loss function:

$$\mathcal{L}_{dec} = \mathcal{L}_{rec} - \delta\mathcal{L}_{gan} \tag{3}$$

where $\delta$ is a regularization coefficient, $\mathcal{L}_{rec} = L_{bce}(\hat{x}, x)$ is a reconstruction loss. The GAN loss is given by $\mathcal{L}_{gan} = \frac{1}{3}[L_{bce}(y_{real}, C_\chi(x)) + L_{bce}(y_{fake}, C_\chi(D_\theta(E_\phi(x)))) + L_{bce}(y_{fake}, C_\chi(D_\theta(z, y)))]$ (Bao et al., 2017), where $y_{real}$ and $y_{fake}$ are vectors of ones and zeros respectively and $z \sim p(z)$. Note that $L_{bce}$ is the binary cross-entropy loss given by $L_{bce}(a, b) = \frac{1}{M}\sum_{i=1}^{M} a\log b + (1-a)\log(1-b)$. The discriminator parameters, $\chi$, are updated to minimize $\mathcal{L}_{gan}$.

The parameters of the encoder, $\phi$, intended for use in synthesizing images from a desired **category**, may be updated by minimizing the following function:

$$\mathcal{L}_{enc} = \mathcal{L}_{rec} + \alpha\mathcal{L}_{KL} + \rho\mathcal{L}_{class} - \delta\mathcal{L}_{gan} \tag{4}$$

where $\alpha$ and $\rho$ are additional regularization coefficients; $\mathcal{L}_{KL} = KL[q_\phi(z|x)||p(z)]$ and $\mathcal{L}_{class} = L_{bce}(\hat{y}, y)$ is the classification loss on the input image. Unfortunately, the loss function in Equation (4) is not sufficient for training an encoder used for attribute manipulation. For this, we propose an additional network and cost function, as described below.

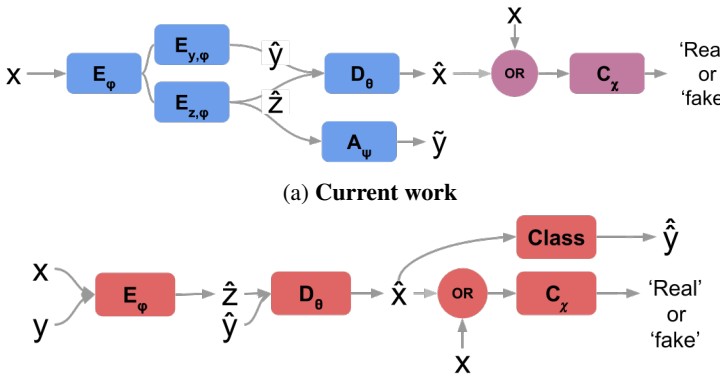

(a) **Current work**

(b) **Previous work** (Bao et al., 2017)

Figure 1: **(a) Current work (adversarial information factorization)** This figure shows our model where the core, shown in blue, is a VAE with information factorization. Note that $E_\phi$ is split in two, $E_{z,\phi}$ and $E_{y,\phi}$, to obtain both a latent encoding, $\hat{z}$, and the label, $\hat{y}$, respectively. $D_\theta$ is the decoder and $A_\psi$ the auxiliary network. The pink blocks show how a GAN architecture may be incorporated by placing a discriminator, $C_\chi$, after the encoder, $E_\phi$, and training $C_\chi$ to classify decoded samples as "fake" and samples from the dataset as "real". For simplicity, the $KL$ regularization is not shown in this figure. **(b) Previous work: cVAE-GAN** (Bao et al., 2017) Architecture most similar to our own. Note that there is no auxiliary network performing information factorization and a label, $\hat{\hat{y}}$, is predicted only for the reconstructed image, rather than for the input image ($\hat{y}$).

## 3.2 ADVERSARIAL INFORMATION FACTORISATION

To factor label information, $y$, out of $\hat{z}$ we introduce an additional auxiliary network, $A_\psi$, that is trained to correctly predict $y$ from $\hat{z}$. The encoder, $E_\phi$, is simultaneously updated to promote $A_\psi$ to make incorrect classifications. In this way, the encoder is encouraged *not* to place attribute information, $y$, in $\hat{z}$. This may be described by the following mini-max objective:

$$\min_\psi \max_\phi L_{bce}(A_\psi(E_{z,\phi}(x)), y) = \min_\psi \max_\phi L_{bce}(\tilde{y}, y) = \min_\psi \max_\phi \mathcal{L}_{aux} \qquad (5)$$

where $E_{z,\phi(x)}$ is the latent output of the encoder.

Training is complete when the auxiliary network, $A_\psi$, is maximally confused and cannot predict $y$ from $\hat{z} = E_{z,\phi}(x)$, where $y$ is the true label of $x$. The encoder loss is therefore given by:

$$\mathcal{L}_{IFcVAE-GAN} = \mathcal{L}_{enc} - \mathcal{L}_{aux} \qquad (6)$$

We call the conditional VAE-GAN trained in this way an Information Factorization cVAE-GAN (IFcVAE-GAN). The training procedure is presented in Algorithm 1.

## 3.3 ATTRIBUTE MANIPULATION

To edit an image such that it has a desired attribute, we encode the image to obtain a $\hat{z}$, the identity representation, append it to our desired attribute label, $\hat{y} \leftarrow y$, and pass this through the decoder. We use $\hat{y} = 0$ and $\hat{y} = 1$ to synthesize samples in each *mode* of the desired attribute e.g. 'Smiling' and 'Not Smiling'. Thus, attribute manipulation becomes a simple 'switch flipping' operation in the representation space.

## 4 RESULTS

In this section, we show both quantitative and qualitative results to evaluate our proposed model. We begin by quantitatively assessing the contribution of adversarial information factorization in an

---

**Algorithm 1 Training Information Factorization cVAE-GAN (IFcVAE-GAN)**: The prior, $p(z) = \mathcal{N}(\mathbf{0}, I)$.

---

1: **procedure** TRAINING CVAE WITH INFORMATION FACTORIZATION
2:      **for** $i$ in $range(N)$ **do**              ▷ $N$ is no. of epochs
3:          ▷ forward pass through all networks
4:          $x \sim \mathcal{D}$             ▷ $\mathcal{D}$ is the training data
5:          $z \sim p(z)$
6:          $\hat{z}, \hat{y} \leftarrow E_\phi(x)$
7:          $\hat{x} \leftarrow D_\theta(\hat{y}, \hat{z})$
8:          $\tilde{y} \leftarrow A_\psi(\hat{z})$          ▷ output of the auxiliary network
9:          # Calculate updates, $u$
10:         # do updates
11:         $\theta \leftarrow \text{RMSprop}(\theta, -\nabla_\theta \mathcal{L}_{dec})$
12:         $\phi \leftarrow \text{RMSprop}(\phi, -\nabla_\phi \mathcal{L}_{IFcVAE-GAN})$      ▷ update the encoder to max. $\mathcal{L}_{aux}$
13:         $\chi \leftarrow \text{RMSprop}(\chi, +\nabla_\chi \mathcal{L}_{gan})$
14:         $\psi \leftarrow \text{RMSprop}(\psi, -\nabla_\psi \mathcal{L}_{aux})$      ▷ update auxiliary network to min. $\mathcal{L}_{aux}$
15:      **end for**
16: **end procedure**

---

ablation study. Following this we perform facial attribute classification using our model. We use a standard deep convolutional GAN, DCGAN, architecture for the ablation study (Radford et al., 2015), and subsequently incorporate residual layers (He et al., 2016) into our model in order to achieve competitive classification results compared with a state of the art model (Zhuang et al., 2018). We finish with a qualitative evaluation of our model, demonstrating how our model may be used for image attribute editing. For our qualitative results we continue to use the same residual networks as those used for classification, since these also improved visual quality.

We refer to any cVAE-GAN that is trained without an $\mathcal{L}_{aux}$ term in the cost function as a naive cVAE-GAN and a cVAE-GAN trained with the $\mathcal{L}_{aux}$ term as an Information Factorization cVAE-GAN (IFcVAE-GAN).

### 4.1 QUANTITATIVE EVALUATION OF (EDITED) IMAGE SAMPLES

When performing image attribute manipulation, there are two important things that we would like to quantify. The first, is reconstruction quality, approximated by the mean squared error, MSE, between $x$ and $\hat{x} = D_\theta(E_{y,\phi(x)}, E_{z,\phi(x)})$.

The second, is the proportion of edited images that have a desired attribute. To approximate this, we train an **independent** classifier on real images to classify the presence ($y = 1$) or absence ($y = 0$) of a desired attribute. We apply the trained classifier to edited images, synthesized using $\hat{y} = 1$ and $\hat{y} = 0$ to obtain classification scores, $\hat{\mathcal{C}}_{Smiling}$ and $\hat{\mathcal{C}}_{Not-Smiling}$ respectively.

### 4.2 QUANTIFYING THE CONTRIBUTION OF ADVERSARIAL INFORMATION FACTORIZATION

Table 1 shows the contributions of each component of our novel function (Equation 6). We consider reconstruction error and classification scores on edited image samples. Smaller reconstruction error indicates better reconstruction, and larger classification scores ($\hat{\mathcal{C}}_{Smiling}$ and $\hat{\mathcal{C}}_{Not-Smiling}$) suggest better control over attribute changes. Note that all input test images for this experiment were from the 'Smiling' category. From Table 1, we make the following observations:

**(1) Our model:** Our model is able to successfully edit an image to have the 'Not Smiling' attribute in 81.3% of cases and the 'Smiling' attribute in all cases.

**(2) Effect of Removing Information Factorization:** Without our proposed $\mathcal{L}_{aux}$ term in the encoder loss function, the model fails completely to perform attribute editing. Since $\hat{\mathcal{C}}_{Smiling}$ + $\hat{\mathcal{C}}_{Not-Smiling} \approx 100\%$, this strongly suggests that samples are synthesized independently of $\hat{y}$ and that the synthesized images are the same for $\hat{y} = 0$ and $\hat{y} = 1$.

**(3) Effect of classifying reconstructed samples:** We explored the effect of including a classification loss on reconstructed samples, $L_{bce}(y, \hat{\hat{y}})$, where $\hat{\hat{y}} = E_{y,\phi}(D_\theta(\hat{x}))$. A similar loss had been proposed by both Bao et al. (2017) and in the GAN literature (Chen et al., 2016; Odena et al., 2016) for conditional image synthesis (rather than attribute editing). To the best of our knowledge, this approach has not been used in the VAE literature. This term is intended to maximise $I(x; y)$ by providing a gradient containing label information to the decoder, however, it does not contribute to the factorization of attribute information, $y$, from $\hat{z}$ and does not provide any clear benefit in our model.

**(4) IcGAN Perarnau et al. (2016):** We choose to include the IcGAN in our ablation study, since it is similar to our model without $\mathcal{L}_{KL}$ and $\mathcal{L}_{aux}$. While the IcGAN achieves a similar reconstruction error to our model it performs less well at attribute editing tasks.

Table 1: **What are the essential parts of the IFcVAE-GAN?** This table shows how novel components of the IFcVAE-GAN loss function affect reconstruction quality, MSE, and the model's ability to edit attributes, $\hat{\mathcal{C}}_{Smiling}$ and $\hat{\mathcal{C}}_{Not-Smiling}$. We used hyper-parameters: $\{\rho = 0.1, \delta = 0.1, \alpha = 0.2\}$. IcGAN reconstruction results are those reported by Lample et al. (2017).
∗ Results obtained by Lample et al. (2017) using humans to evaluate the images.

| Model | MSE | $\hat{\mathcal{C}}_{Not-Smiling}$ | $\hat{\mathcal{C}}_{Smiling}$ |
|---|---|---|---|
| (1) Ours | 0.028 | 81.3% | 100.0% |
| (2) Without $\mathcal{L}_{aux}$ | 0.028 | 18.8% | 81.3% |
| (3) With classification loss on $\hat{x}$ | 0.028 | 93.8% | 93.8% |
| (4) IcGAN (Perarnau et al., 2016) | 0.028 | 9.9∗% | 91.9∗% |

## 4.3 FACIAL ATTRIBUTE CLASSIFICATION

We have proposed a model that learns a representation, $\{\hat{z}, \hat{y}\}$, for faces such that the identity of the person, encoded in $\hat{z}$, is factored from a particular facial attribute. We achieve this by minimizing the mutual information between the identity encoding and the facial attribute encoding to ensure that $H(y|\hat{z}) \approx H(y)$, while also training $E_{y,\phi}$ as an attribute classifier. Our training procedure encourages the model to put all label information into $\hat{y}$, rather than $\hat{z}$. This suggests that our model may be useful for facial attribute classification.

To further illustrate that our model is able to separate the representation of particular attributes from the representation of the person's identity, we can measure the model's ability, specifically the encoder, to classify facial attributes. We proceed to use $E_{y,\phi}$ directly for facial attribute classification and compare the performance of our model to that of a state of the art classifier proposed by Zhuang et al. (2018). Results in Figure 2 show that our model is highly competitive with a state of the art facial attribute classifier Zhuang et al. (2018). We outperformed by more than 1% on 2 out of 10 categories, underperformed by more than 1% on only 1 category and remained competitive with all other attributes. These results demonstrate that the model is effectively factorizing out information about the attribute from the identity representation.

## 4.4 QUALITATIVE RESULTS

In this section, we focus on attribute manipulation (described previously in Section 3.3). Briefly, this involves reconstructing an input image, $x$, for different attribute values, $\hat{y} \in \{0, 1\}$.

We begin by demonstrating how a cVAE-GAN (Bao et al. Bao et al. (2017)) may fail to edit desired attributes, particularly when it is trained to achieve low reconstruction error. The work of Bao et al. Bao et al. (2017) focused solely on the ability to synthesise images with a desired attribute, rather than to reconstruct a particular image and edit a specific attribute. It is challenging to learn a representation that both preserves identity and allows factorisation Higgins et al. (2016). Figure 3(c,e) shows edited images, setting $\hat{y} = 0$ for 'Not Smiling' and $\hat{y} = 1$ for 'Smiling'. We found that the cVAE-GAN (Bao et al. Bao et al. (2017)) failed to edit samples for the $y = 0$ ('Not Smiling') case. This failure demonstrates the need for models that learn a factored latent representation, while maintaining good reconstruction quality. Note that we achieve good reconstruction by reducing

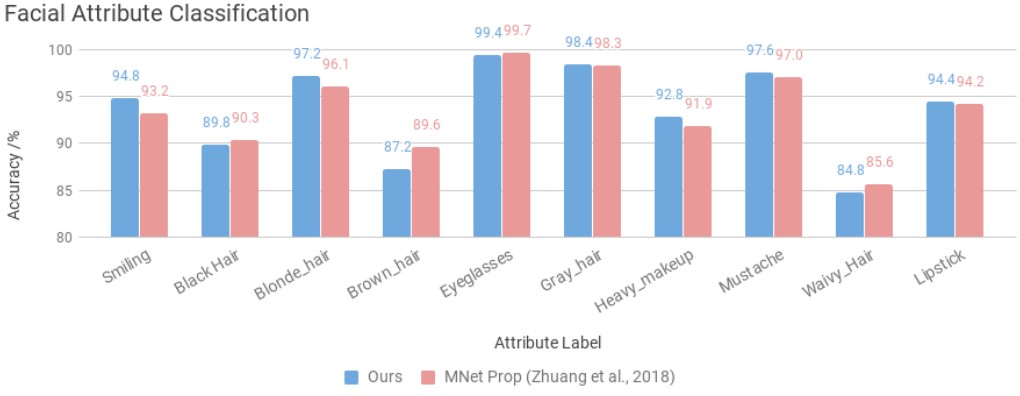

Figure 2: **Facial Attribute Classification.** We compare the performance of our classifier, $E_{y,\phi}$, to a state of art classifier (Zhuang et al., 2018).

weightings on the $KL$ and GAN loss terms, using $\alpha = 0.005$ and $\delta = 0.005$ respectively. We trained the model using RMSProp Tieleman & Hinton (2012) with momentum $= 0.5$ in the discriminator.

We train our proposed IFcVAE-GAN model using the same optimiser and hyper-parameters that were used for the Bao et al. (2017) model above. We also used the same number of layers (and residual layers) in our encoder, decoder and discriminator networks as those used by Bao et al. (2017). Under this set-up, we used the following additional hyper-parameter: $\{\rho = 1.0\}$ in our model. Figure 3 shows reconstructions when setting $\hat{y} = 0$ for 'Not Smiling' and $\hat{y} = 1$ for 'Smiling'. In contrast to the naive cVAE-GAN (Bao et al., 2017), our model is able to achieve good reconstruction, capturing the identity of the person, while also being able to change the desired attribute. Table 2 shows that the model was able to synthesize images with the 'Not Smiling' attribute with a 98% success rate, compared with a 22% success rate using the naive cVAE-GAN Bao et al. (2017).

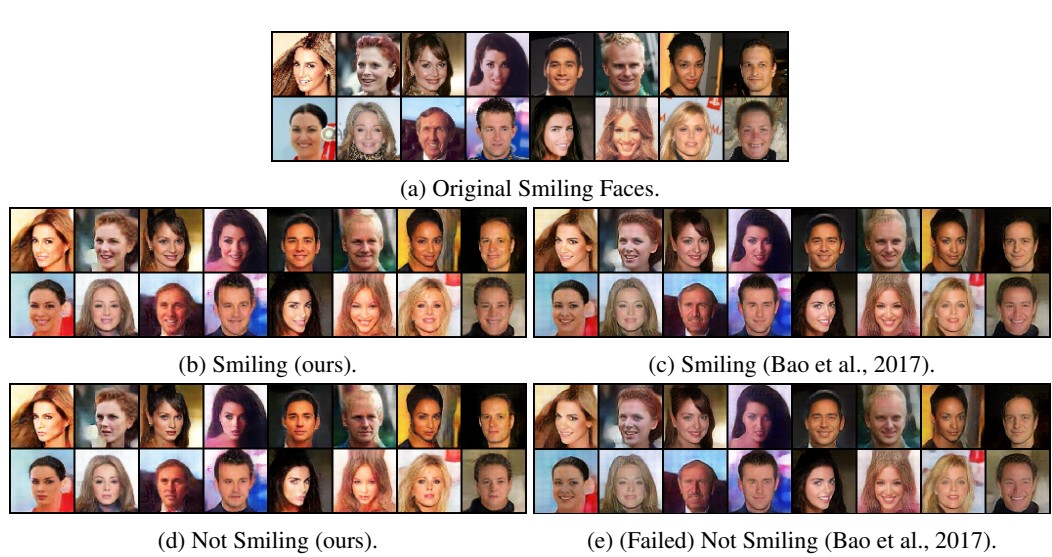

Figure 3: **Reconstructions, 'Smiling' and 'Not Smiling'.** The goal here was to reconstruct the face, changing only the desired 'Smiling' attribute. This demonstrates how other conditional models (Bao et al., 2017) may fail at the image attribute editing task, when high quality reconstructions are required. Both models are trained with the same optimizers and hyper-parameters.

Table 2: **Comparing our model, the IFcVAE-GAN, to the naive cVAE-GAN Bao et al. (2017)**. $\hat{\mathcal{C}}_{Smiling}$ and $\hat{\mathcal{C}}_{Not-Smiling}$ denote the proportion of edited samples that have the desired attribute. We see that both models achieve comparable (MSE) reconstruction errors, however, only our model is able to synthesize images of faces without smiles. A complete ablation study for this model (with residual layers) is given in the appendix (Table 3).

| Model | MSE | $\hat{\mathcal{C}}_{Not-Smiling}$ | $\hat{\mathcal{C}}_{Smiling}$ |
|---|---|---|---|
| Ours (with residual layers) | 0.011 | 98% | 100% |
| cVAE-GAN Bao et al. (2017) (with residual layers) | 0.011 | 22% | 85% |

## 4.5 EDITING OTHER FACIAL ATTRIBUTES

In this section we apply our proposed method to manipulate other facial attributes where the initial samples, from which the $\hat{z}$'s are obtained, are test samples whose labels are $y = 1$ indicating the presence of the desired attribute (e.g. 'Blonde Hair'). In Figure 4, we observe that our model is able to both achieve high quality reconstruction and edit the desired attributes.

$\hat{y} = 0$

$\hat{y} = 1$

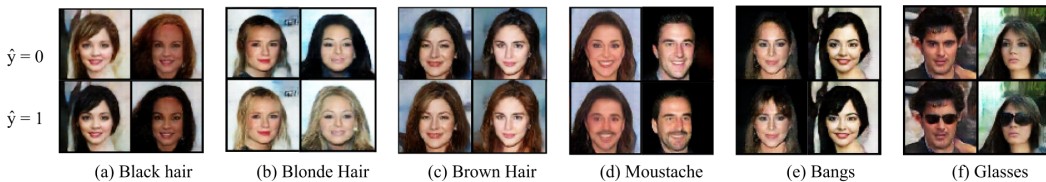

(a) Black hair     (b) Blonde Hair     (c) Brown Hair     (d) Moustache     (e) Bangs     (f) Glasses

Figure 4: **Editing other attributes.** We obtain a $\hat{z}$, the identity representation, by passing an image, $x$ through the encoder. We append $\hat{z}$ with a desired attribute label, $\hat{y} \leftarrow y$, and pass this through the decoder. We use $\hat{y} = 0$ and $\hat{y} = 1$ to synthesize samples in each *mode* of the desired attribute

We have presented the novel IFcVAE-GAN model, and (1) demonstrated that our model learns to factor attributes from identity, (2) performed an ablation study to highlight the benefits of using an auxiliary classifier to factorize the representation and (3) shown that our model may be used to achieve competitive scores on a facial attribute classification task. We now discuss this work in the context of other related approaches.

## 5 COMPARISON TO RELATED WORK

We have used adversarial training (involving an auxiliary classifier) to factor attribute label information, $y$, out of the encoded latent representation, $\hat{z}$. Schmidhuber (2008) and Lample et al. (2017) perform similar factorization of the latent space. Similarly to us, Lample et al. (2017) incorporate this factorisation technique into the encoder of a generative model, however, unlike in our model, their encoder does not predict attribute information and so may not be used as a classifier.

Belghazi et al. (2018) proposed a general approach for predicting the mutual information, which may then be minimized via an additional model. Rather than predicting mutual information (Belghazi et al., 2018) between latent representations and labels, we implicitly minimize it via adversarial information factorization.

Our work has the closest resemblance to the cVAE-GAN architecture (see Figure 1) proposed by Bao et al. (2017). cVAE-GAN is designed for synthesizing samples of a particular class, rather than manipulating a single attribute of an image from a class. In short, their objective is to synthesize a "Hathway" face, whereas our objective would be to make "Hathway smiling" or "Hathway not smiling", which has different demands on the type of factorization in the latent representation. Separating categories is a simpler problem since it is possible to have distinct categories and changing categories may result in more noticeable changes in the image. Changing an attribute requires a specific and targeted change with minimal changes to the rest of the image. Additionally, our model simultaneously learns a classifier for input images unlike the work by Bao et al. (2017).

In a similar vein to our work, Antipov et al. (2017) acknowledge the need for "identity preservation" in the latent space. They achieve this by introducing an identity classification loss between an input data sample and a reconstructed data sample, rather than trying to separate information in the encoding itself. Similar to our work, Larsen et al. (2016) use a VAE-GAN architecture. However, they do not condition on label information and their image "editing" process is not done in an end-to-end fashion (likewise with Upchurch et al. (2017)).

Our work highlights an important difference between category conditional image synthesis (Bao et al., 2017) and attribute editing in images (Lample et al., 2017; Perarnau et al., 2016): what works for category conditional image synthesis may not work for attribute editing. Furthermore, we have shown (Section 4.2) that for attribute editing to be successful, it is necessary to factor label information out of the latent encoding.

In this paper, we have focused on latent space generative models, where a small change in latent space results in a semantically meaningful change in image space. Our approach is orthogonal to a class of image *editing* models, called "image-to-image" models, which aim to learn a single latent representation for images in different domains. Recently, there has been progress in image-to-image domain adaptation, whereby an image is translated from one domain (e.g. a photograph of a scene) to another domain (e.g. a painting of a similar scene) (Zhu et al., 2017; Liu et al., 2017; Liu & Tuzel, 2016). Image-to-image methods may be used to translate smiling faces to non-smiling faces, however, these models (Liu et al., 2017; Liu & Tuzel, 2016) require significantly more resources than ours[2]. By performing factorization in the latent space, we are able to use a single generative model, to edit an attribute by simply changing a single unit of the encoding, $y$, from 0 to 1 or vice versa.

Finally, while we use labelled data to learn representations, we acknowledge that there are many other models that learn factored, or disentangled, representations from unlabelled data including several VAE variants (Higgins et al., 2016; Burgess et al., 2018; Kumar et al., 2018). The $\beta$-VAE (Higgins et al., 2016) objective is similar to the information bottleneck (Burgess et al., 2018), minimizing mutual information, $I(x; z)$, which forces the model to exploit regularities in the data and learn a disentangled representation. In our approach we perform a more direct, supervised, factorisation of the latent space, using a mini-max objective, which has the effect of approximately minimizing $I(z; y)$.

## 6 CONCLUSION

We have proposed a novel perspective and approach to learning representations of images which subsequently allows elements, or attributes, of the image to be modified. We have demonstrated our approach on images of the human face, however, the method is generalisable to other objects. We modelled a human face in two parts, with a continuous latent vector that captures the identity of a person and a binary unit vector that captures a facial attribute, such as whether or not a person is smiling. By modelling an image with two separate representations, one for the object and the other for the object's attribute, we are able to change attributes without affecting the identity of the object. To learn this factored representation we have proposed a novel model aptly named Information Factorization conditional VAE-GAN. The model encourages the attribute information to be factored out of the identity representation via an adversarial learning process. Crucially, the representation learned by our model **both** captures identity faithfully and facilitates accurate and easy attribute editing without affecting identity. We have demonstrated that our model performs better than pre-existing models intended for category conditional image synthesis (Section 4.4), and have performed a detailed ablation study (Table 1) which confirms the importance and relevance of our proposed method. Indeed, our model is highly effective as a classifier, achieving state of the art accuracy on facial attribute classification for several attributes (Figure 2). Our approach to learning factored representations for images is both a novel and important contribution to the general field of representation learning.

---

[2]While our approach requires a single generative model, the approaches of Liu et al. (2017); Liu & Tuzel (2016) require a pair of generator networks, one for each domain.

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

Table 3: **What are the essential parts of the IFcVAE-GAN (with residual layers)?** This table shows how novel components of the IFcVAE-GAN loss function affect mean squared (reconstruction) error, MSE, and the proportion of edited samples with the desires attribute, $\hat{\mathcal{C}}_{Smiling}$ and $\hat{\mathcal{C}}_{Not-Smiling}$. We use hyper-parameters: $\{\rho = 0.1, \delta = 0.005, \alpha = 0.005, momentum = 0.5\}$. We also show classification accuracy (Acc.) of $E_{y,\phi}$. IcGAN reconstruction results are those reported by Lample et al. (2017).
*Note that the model of Bao et al. (2017) does not incorporate a classifier. **Results obtained by Lample et al. (2017) using humans to evaluate the images.

| Model | MSE | $\hat{\mathcal{C}}_{Not-Smiling}$ | $\hat{\mathcal{C}}_{Smiling}$ | Acc. ($E_{y,\phi}$) |
|---|---|---|---|---|
| Ours (with residual layers) ($\alpha = 0.005$) | 0.011 | 98% | 100.0% | 92% |
| Higher levels of regularization ($\alpha = 0.1$) | 0.020 | 100% | 100% | 92% |
| Without $\mathcal{L}_{aux}$, ($\alpha = 0.005$) | 0.013 | 28% | 91% | 91% |
| Without $\mathcal{L}_{KL}$, ($\alpha = 0.005$) | 0.039 | 100% | 96% | 90% |
| Without $\mathcal{L}_{gan}$, ($\alpha = 0.005$) | 0.009 | 100% | 100% | 88% |
| cVAE-GAN Bao et al. (2017), ($\alpha = 0.005$) | 0.011 | 22% | 85% | n/a* |
| IcGAN (Perarnau et al., 2016) | 0.028 | 9.9**% | 91.9**% | n/a |

Jun-Yan Zhu, Philipp Krähenbühl, Eli Shechtman, and Alexei A Efros. Generative visual manipulation on the natural image manifold. In *European Conference on Computer Vision*, pp. 597–613. Springer, 2016.

Jun-Yan Zhu, Taesung Park, Phillip Isola, and Alexei A Efros. Unpaired image-to-image translation using cycle-consistent adversarial networks. *arXiv preprint arXiv:1703.10593*, 2017.

Ni Zhuang, Yan Yan, Si Chen, Hanzi Wang, and Chunhua Shen. Multi-label learning based deep transfer neural network for facial attribute classification. *Pattern Recognition*, 80:225–240, 2018.

# APPENDIX

## ABLATION STUDY FOR OUR MODEL WITH RESIDUAL LAYERS

For completeness we include a table (Table 3) demonstrating an ablation study for our model with the residual network architecture discussed in Section 4.4, note that this is the same architecture that was used by Bao et al. (2017). Table 3 and additionally, Figure 5, demonstrate the need for the $\mathcal{L}_{aux}$ loss and shows that increased regularisation reduces reconstruction quality. The table also shows that there is no significant benefit to using the $\hat{\mathcal{L}}_{class}$ loss. These findings are consistent with those of the ablation study in the main body of the text for the IFcVAE-GAN with a the GAN architecture of Radford et al. (2015).

We additionally show results without $\mathcal{L}_{KL}$ and $\mathcal{L}_{gan}$. Results show that small amounts of $KL$ regularisation are required to achieve good reconstruction. Models trained without $\mathcal{L}_{gan}$ achieve slightly lower reconstruction error than other models, however, the reconstructed images are blurred (see Figure 6). Interestingly, when our model is trained without the $\mathcal{L}_{gan}$ or $\mathcal{L}_{KL}$ loss, it is still able to edit attributes with high accuracy, however, the visual quality of samples is poor. This shows that the attribute information is still factored from the rest of the latent representation, which is the main contribution of our work.

## ADDITIONAL CLASSIFICATION PERFORMANCE COMPARISON WITH A MODEL TRAINED TO PERFORM DISENTANGLEMENT

In our model we use labelled data to learn factored representations, however, there are many other models that learn factored, or disentangled, representations from unlabelled data including several variational autoencoder (Rezende et al., 2014; Kingma & Welling, 2013) variants (Higgins et al., 2016; Burgess et al., 2018; Kumar et al., 2018). Once trained, the representation learned by each of these models may be evaluated by training a linear classifier on latent encodings (Kumar et al., 2018;

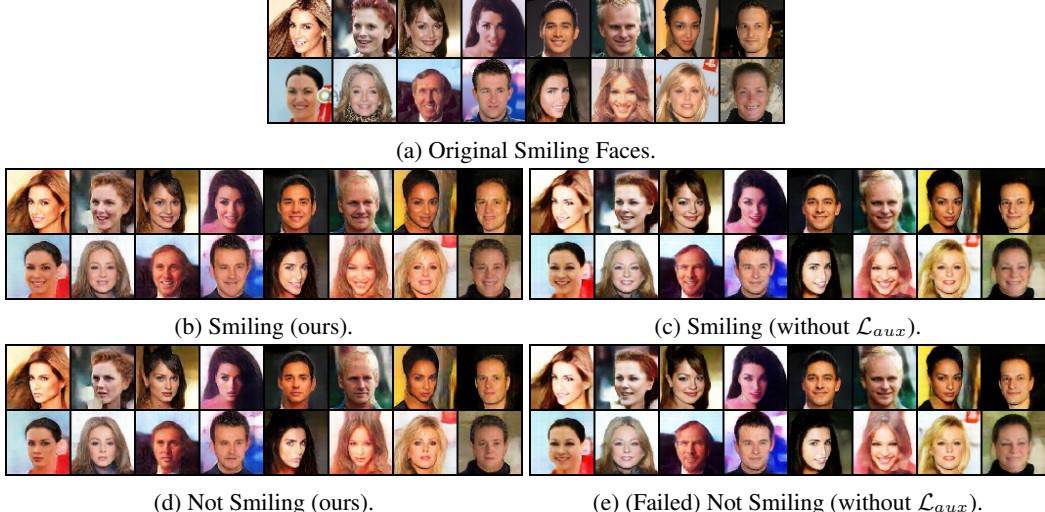

Figure 5: **Reconstructions, 'Smiling' and 'Not Smiling', with and without $\mathcal{L}_{aux}$.** The goal here was to reconstruct the face, changing only the desired 'Smiling' attribute. This figure demonstrates the need for the $\mathcal{L}_{aux}$ term in our model. Both models are trained with the same optimizers and hyper-parameters.

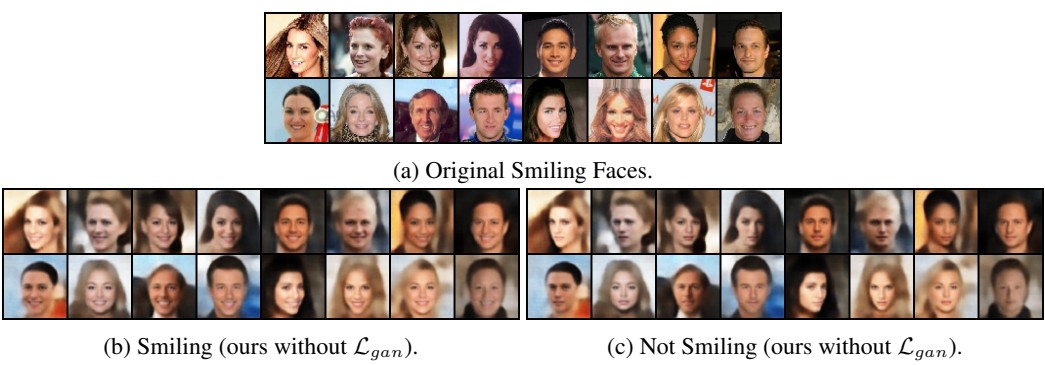

Figure 6: **Reconstructions, 'Smiling' and 'Not Smiling', without $\mathcal{L}_{gan}$.** This figure demonstrates that without $\mathcal{L}_{gan}$, reconstructions are blurred. To achieve sharp reconstructions it is necessary to incorporate $\mathcal{L}_{gan}$ in the loss function.

Higgins et al., 2016). Figure 7 shows classification results achieved using our model compared to one of the best models (Kumar et al., 2018) proposed for learning disentangled representations.

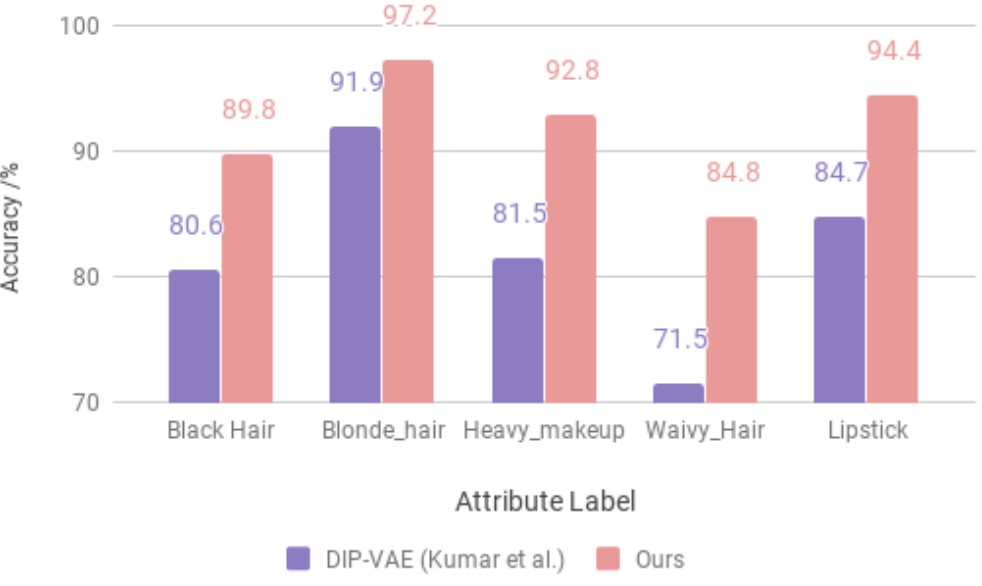

Figure 7: **Facial Attribute Classification.** We compare the performance of our classifier, $E_{y,\phi}$, to a linear classifier trained on latent representations extracted from a trained DIP-VAE (Kumar et al., 2018). DIP-VAE is one of the best models for learning disentangled (, or factored,) representations from unlabelled data.

