# OpenReview forum: "Adversarial Information Factorization"
_ICLR.cc/2019/Conference_

### Official Review · AnonReviewer3 · 2018-10-31
**Interesting idea, Too complex model**

**Rating:** 6
**Confidence:** 4

**Review:**

This paper proposed a generative model to learn the representation which can separates the identity of an object from an attribute. Authors extended the autoencoder adversarial by adding an auxiliary network.

Strength
The motivation of adding this auxiliary network, which is to distinguish the information between latent code z and attribute vector y, is clean and clear.
Experiments illustrate the advantage of using auxiliary network and demonstrating the role of classify. Experimental results also show the proposed model learning to factor attributes from identity on the face dataset.

Weakness
The proposed model seem to be unnecessarily complex. For example, the loss of  in (6) actually includes 6 components (5 are from L_enc) and 4~5 tuning hyper-parameters. The L_gan also includes 3 parts. The reason of adding gan loss lacks either theoretical or empirical analysis. So as L_KL. In addition, the second term in L_gan is unnecessary since you already have a reconstruction loss. It also make it to be unclear what we obtain if the equilibrium of the GAN objective achieved.

The written of this paper can be improved to make it more clear.
It looks \hat_y and \tilde_y are same thing.
How do you get \hat_z? Do you assume the posterior distribution is Gaussian and use the reparameterization trick? What are \hat_y and \hat_\hat_y? Are they binary or a scalar between 0 and 1?  How do you generate \hat_x? When generating \hat_x, do you sample \hat_z and \hat_y? If so, how do treat the variance problem of \hat_y?

---

> ### Author Response · Authors · 2018-11-12
> **Our model is no more complex than others.**
>
> The reviewer's main concern is that the model is too complex, however, our proposed model is no more complex than the accepted paper of Bao et al. Our cost has the same number of components and hyper parameters and our model has the same number of networks (our encoder network has two outputs). Most of our components are also less complex because losses are computed on network outputs rather than on features extracted from multiple intermediate layers. Additionally, we demonstrate that terms in our loss function, \hat{L}_{class}, may be excluded, making our model less complex.
>
> Throughout our work we have been intentionally explicit and detailed about the costs we use. This may have resulted in the complexity of our approach being excessively emphasised, however, it is merely a thorough presentation of our idea. If complexity is the reviewer’s main concern, our paper is no more complex than papers previously accepted. If this is the main criticism, our paper should be accepted.
>
> We would again like to thank the reviewer for their constructive feedback, which has enabled us to improve our paper.

---

> ### Author Response · Authors · 2018-11-12
> **Addressing questions of reviewer 3**
>
> [Reviewer]
> The written of this paper can be improved to make it more clear.
> It looks \hat_y and \tilde_y are same thing.
>
> [Authors]
> Line 6 of Algorithm 1 (as well as Figure 1) show that \hat{y} is one of the outputs of the encoder. This is also described in Section 2.4: “the encoder outputs both a latent vector, \hat{z}, and an attribute vector, \hat{y}”
>
> Line 8 of Algorithm 1 (as well as Figure 1) show that \tilde{y} is the output of the auxiliary classifier. This is also described in Section 3.1: "We introduce an auxiliary network, A_\psi : \hat{z} —> \tilde{y},”
>
> To re-iterate, \hat{y} is one of the outputs of the encoder. \tilde{y} is the output of the auxiliary classifier. They are not the same thing.
>
> [Reviewer]
> How do you get \hat_z? Do you assume the posterior distribution is Gaussian and use the reparameterization trick?
>
> [Authors]
> Yes, we assume a Gaussian prior and posterior and use the re-parametrization trick. On page 4 we say “We integrate this novel factorisation method into a VAE-GAN.” And we describe VAE-GANs in Section 2. Specifically, we describe the re-parametrization trick in Section 2.1:
>
> "The encoder predicts, \mu_\phi(x) and \sigma_\phi(x) for a given input x and a latent sample, \hat{z}, is drawn from q_\phi(z|x) as follows: \epsilon \sim \mathcal{N}(\mathbf{0},I) then z = \mu_\phi(x) + \sigma_\phi(x) \odot \epsilon."
>
> There was a typo here, the paper now reads: “then \hat{z} = \mu_\phi(x) + \sigma_\phi(x)”
>
> [Reviewer]
> What are \hat_y and \hat_\hat_y? Are they binary or a scalar between 0 and 1?
>
> [Authors]
> Line 6 of Algorithm 1 (as well as Figure 1) show that \hat{y} is one of the outputs of the encoder.
>
> Line 8 of Algorithm 1 (as well as Figure 1) show that \hat\hat{y} is the “predicted label of a reconstructed data sample”.
>
> \hat_y and \hat_\hat_y are continuous scalars between [0,1]. We have now added in Section 2.5:  “\hat{y} \in [0,1]” and in Section 3.1 we have added “\hat{\hat{y}} \in [0,1]” to make this more clear.
>
> Thank you for helping us to improve our paper with this comment.
>
> [Reviewer]
> How do you generate \hat_x? When generating \hat_x, do you sample \hat_z and \hat_y? If so, how do treat the variance problem of \hat_y?
>
> [Authors]
> To obtain \hat{x} we pass \hat{z} and \hat{y} through the decoder, D_\theta. During training \hat{z} and \hat{y} are the output of the encoder, which takes a sample from the data as input. During attribute manipulation, \hat{z} is still the output of the encoder, and we set \hat{y} to 0 or 1. This is detailed in the paper as follows:
>
> During training:
> Lines 4, 6, and 7 of Algorithm 1 show how \hat_x is synthesised.
> Line 4: An image x is sampled from the data
> Line 6: x is passed through the encoder, which outputs \hat{z} and \hat{y}.
> Line 7: \hat{z} and \hat{y} are concatenated and passed to the decoder.
>
> During attribute manipulation:
> Quote taken from Section 3.3 of our paper:
> we encode the image to obtain a \hat{z}, the identity representation, append it to our desired attribute label, \hat{y} <— y, and pass this through the decoder. We use \hat{y}=0 and \hat{y}=1 to synthesize samples in each mode of the desired attribute e.g. `Smiling' and `Not Smiling'.

---

> ### Author Response · Authors · 2018-11-12
> **We have addressed comments from reviewer 3**
>
> [Reviewer]
> Weakness
> The proposed model seem to be unnecessarily complex. For example, the loss of in (6) actually includes 6 components (5 are from L_enc) and 4~5 tuning hyper-parameters.
>
> [Authors]
> We appreciate that our model has several components, however, the original Bao et al. model also consists of 6 components (see Equation 7 of Bao et al.) along with 4 hyper-parameters. Since our model is no more complex than that of Bao et al. — an accepted paper — we assert that this should not be seen as a weakness of our paper.
>
> Despite having a few hyper-parameters, our model does not require extensive hyper-parameter tuning. However, to obtain high fidelity reconstructions it is necessary to select low values of delta (the weight on the L_gan term) and alpha (the weight on the KL term).
>
> [Reviewer]
> The L_gan also includes 3 parts.
>
> [Authors]
> The L_gan term is similar to the one used by Bao et al, which also has three components. Note also that the GAN loss is intended purely to improve the visual quality of the samples. Our contribution is still valid without L_gan, since our main contribution is the introduction of an auxiliary network, A_\psi, and the loss, L_aux.
>
> [Reviewer]
> The reason of adding gan loss lacks either theoretical or empirical analysis. So as L_KL.
>
> [Authors]
> The use of the gan loss and the KL loss are motivated already by Bao et al. in the VAE-GAN which we introduce in Section 2.
>
> According to our understanding of the GAN literature, it is generally accepted that GAN loss improves the visual quality of samples. Experimentally, we found this to be true for our model also.  Similarly, regularisation of the latent space helps with generalisation to test samples. We found that our model performed better with a small amount of KL regularisation than without any, and that models trained without regularisation overfit and had very poor reconstruction (MSE=0.0381).
>
> Interestingly, when our model is trained without a GAN loss or KL loss, it is still able to edit attributes with high accuracy, however, the visual quality of samples is poor. This shows that the attribute information is still factored from the rest of the latent representation, which is the main contribution of our work.
>
> [Reviewer]
> In addition, the second term in L_gan is unnecessary since you already have a reconstruction loss. It also make it to be unclear what we obtain if the equilibrium of the GAN objective achieved.
>
> [Authors]
> We use a similar L_gan to that used by Bao et al. Please refer to Algorithm 1, line 9 of Bao et al. In the cVAE-GAN there are two sources of `fake' images, (a) reconstructed images, D_\theta(E_\phi(x)) and (b) sampled images, D_\theta(z), z ~ p(z), where p(z) is the prior. This is why there are three terms in the GAN loss. Reconstruction loss alone is often not enough to achieve high quality reconstruction.
>
> We explicitly wrote out the GAN loss as three terms for clarity, but the GAN loss could still be written as two terms, where C_\chi is the discriminator:
> E_p_real(x) log C_\chi(x) + E_p_fake(x) log (1 - C_\chi(x)), where sampling x_fake ~ p_fake(x), includes both reconstructed images and sampled images. This is the same as the original GAN loss.

---

> > ### Comment · AnonReviewer3 · 2018-11-12
> > **thanks for clarifying**
> >
> > 1) While these components(e.g., L_gan and L_KL) are verified in Bao et al., it is still possible that they are not necessary in this model since you add some new staff. Since you have the experiments which demonstrates the importance of adding these staffs, can you add the results to clarify this or at least add several sentences to mention this?
> >
> > 2) In the original GAN, if the equilibrium of min-max objective is achieved, we will have p_data = p_model. Is there anything similar in your model? What will we obtain if the equilibrium of your min-max objective is achieved? This part seems to be not very clear and make your method to be not that "principle".

---

> > > ### Author Response · Authors · 2018-11-13
> > > **Added results for our model trained without L_gan or L_KL**
> > >
> > > [Reviewer]
> > > 1) While these components(e.g., L_gan and L_KL) are verified in Bao et al., it is still possible that they are not necessary in this model since you add some new staff. Since you have the experiments which demonstrates the importance of adding these staffs, can you add the results to clarify this or at least add several sentences to mention this?
> > >
> > > [Authors]
> > > Thank you for the suggestion. We have updated Table 3 to include results for our model trained without L_gan and without L_KL and added additional text to the appendix. We have also added Figure 6 which demonstrates the blurred images obtained if the GAN loss is not used.
> > >
> > >
> > > [Reviewer]
> > > 2) In the original GAN, if the equilibrium of min-max objective is achieved, we will have p_data = p_model. Is there anything similar in your model? What will we obtain if the equilibrium of your min-max objective is achieved? This part seems to be not very clear and make your method to be not that "principle".
> > >
> > > [Authors]
> > >
> > > The objective function as we have written it above, E_p_real(x) log C_\chi(x) + E_p_fake(x) log (1 - C_\chi(x)), is in the exact same form as the original objective, simply with different notation. In our case, we refer to p_data and p_model as p_real and p_fake respectively. C_\chi is the discriminator. Therefore, if the generator (in our case the decoder) and the discriminator are optimal, it follows that p_real = p_fake. Recall (from above) p_fake is the distribution of reconstructed and synthesised images (p_model) and p_real are samples from the (training) data (p_data). Therefore, when optimal the reconstructed and synthesised samples appear to come from the same distribution as the training data.

---

> ### Author Response · Authors · 2018-11-12
> **Thank you for acknowledging that our paper is `clear' and `interesting'**
>
> [Reviewer]
> This paper proposed a generative model to learn the representation which can separates the identity of an object from an attribute. Authors extended the autoencoder adversarial by adding an auxiliary network.
>
> [Reviewer]
> Strength
> The motivation of adding this auxiliary network, which is to distinguish the information between latent code z and attribute vector y, is clean and clear.
> Experiments illustrate the advantage of using auxiliary network and demonstrating the role of classify. Experimental results also show the proposed model learning to factor attributes from identity on the face dataset.
>
> [Authors]
> We thank the reviewer for acknowledging this paper is “clear”, “interesting” and that experiments presented in our paper do indeed support our proposed method. The reviewer appears to have a very good understanding of the contributions made in our paper.

---

### Official Review · AnonReviewer2 · 2018-11-01
**missing references to previous work**

**Rating:** 6
**Confidence:** 4

**Review:**

Summary:

This paper builds upon the work of Boa et al (2017 ) (Conditional VAE GAN) to allow attribute manipulation in the synthesis process.

In order to disentangle the identity information from the attributes the paper proposes adversarial information factorization : let z be the latent code and y be the attribute the paper proposes to have p(y) =  p(y|z= E_phi(x)), i.e to have z independent of y.  This disentanglement is implemented through a GAN on the variable y  min _phi Distance (p(y), p(y|z)), the distance is defined via a discriminator on y.

Experiments are presented on celeba dataset,  1) on attribute manipulation from smiling to non smiling for example, on 2) attribute classification results are presented , 3) ablation studies are given to study the effect of each component of the model highlighting the effect of the adversarial information factorization.

Originality Novelty:

There is a large body of work on disentanglement that the paper does not cite or compare to for instance, InfoGAN,  Beta- VAE https://openreview.net/pdf?id=Sy2fzU9gl and disentangled latent concepts https://arxiv.org/pdf/1711.00848.pdf

Note that for example that in beta- VAE it is a similar idea where but it is on z and z|x and the distance used is KL (since it is has closed form with gaussian) , min_phi Loss+ beta KL (p(z), p(z|x)), a discussion of the previous related work in the paper is necessary.

The work is also related to MINE https://arxiv.org/pdf/1801.04062.pdf where one would like to minimize the mutual information I(z;y)  this mutual information is estimated through a min/max game.

Questions:

-  why is RMSprop used for optimization, your model and the Bao et al baseline might benefit from the use of Adam?

- (Table 3 in appendix ) Have you tried higher values of alpha the weight of KL, with the model of Bao et al (it is recommended in beta VAE to have high value of what you call alpha)?

Overall assessment:

The paper novelty is using min/max game to estimate the mutual information between y (attribute) and z (identity code). Disentanglement and use of min/max games for estimating mutual information has been explored before.  Further discussion and comparaison to previous work is needed.

---

> ### Author Response · Authors · 2018-11-09
> **Improved Related Work.**
>
> [Reviewer]
> Overall assessment:
> The paper novelty is using min/max game to estimate the mutual information between y (attribute) and z (identity code). Disentanglement and use of min/max games for estimating mutual information has been explored before.  Further discussion and comparison to previous work is needed.
>
> [Authors]
> As mentioned above, to the best of our knowledge, we cannot find the implied connection of MINE with performing an explicit mini-max game, which our paper proposes. We would appreciate it if the reviewer could please let us know what they had in mind when making this connection?
>
> We appreciate the additional references that the reviewer proposed, two of which we had already included (beta-VAE and InfoGAN). Based on these very helpful and constructive suggestions we have improved the the related work section of our paper, by adding the following:
>
> "Finally, while we use labelled data to learn representations, we acknowledge that there are many other models that learn factored, or disentangled, representations from unlabelled data including several VAE variants \citep{higgins2016beta, kumar2018variational}. The beta-VAE \cite{higgins2016beta} objective is similar to the information bottleneck \cite{burgess2018understanding}, minimizing mutual information, I(x;z), which forces the model to exploit regularities in the data and learn a disentangled representation. In our approach we perform a more direct, supervised, factorisation of the latent space, using a mini-max objective, which has the effect of approximately minimizing I(z;y)."
>
> We agree that disentanglement has indeed been studied before, however, when making comparisons we have focused on comparing to models that, like ours, make use of labelled data. When comparing our model to previous work, we chose the most challenging benchmark for facial attribute classification, not just those that use disentangled representations. Those that use disentangled representations perform worse than this benchmark. Our classification results are highly competitive with this benchmark.
>
> To the best of our knowledge, our work is the * only * approach to learn disentangled representations which enable image attribute manipulation and simultaneously achieves competitive results with state of the art models on image classification. We believe the demonstrated versatility and novelty of this work are strong grounds for acceptance.
>
> Again, we would like to sincerely thank the reviewer for helping us to improve our paper with their constructive suggestions.

---

> > ### Comment · AnonReviewer2 · 2018-12-05
> > **clarification regarding MINE**
> >
> > I thank the authors for their revision and for addressing my concerns.
> >
> > I maybe did not express myself well , sorry for the confusion. Yes of course estimating mutual information in MINE does not use a min/max game, one estimate the mutual information with the variational form and then if the goal is to minimize this mutual information to learn an auxiliary network, one would get the min/max game.
> >
> > What I meant regarding mine is that  your approach is to work on making the distance between
> >  P(tilde{y}= A_{psi}(z)|z=E_phi(x))  and P(y) the smallest possible. your approach is to say we  optimize on the A_{psi} to predict correctly and on E_{phi} to make errors.    The only thing is that quantity is intuitive but it is not immediately linked explicitly to a mutual information estimation. If you have any formal insight how your approach would link to that that would be great addition to the paper.
> >
> > Another approach would have been using MINE:
> >
> > min_{phi} max_{T}    E_{(x,y) joint }T(E_{phi}(x), y) - log (E_{x,y random } T(E_{phi}(x), y) )
> >
> > this would be l_aux that one would add.
> >
> > I am not asking to baseline this now in the current paper - I acknowledge the novelty of your proposal - but this would have been much more linked to  information based factorization as the title suggests.

---

> ### Author Response · Authors · 2018-11-09
> **Addressing Questions Of Reviewer 2**
>
> [Reviewer]
> Questions:
> -  why is RMSprop used for optimization, your model and the Bao et al baseline might benefit from the use of Adam?
>
> [Authors]
> We experimented with both RMSprop and Adam for both the Bao et al model and ours and generally found RMSprop to give better quality reconstructions.
>
> [Reviewer]
> - (Table 3 in appendix ) Have you tried higher values of alpha the weight of KL, with the model of Bao et al (it is recommended in beta VAE to have high value of what you call alpha)?
>
> [Authors]
> As discussed in the beta-VAE paper, higher beta values (in our case alpha) often lead to worse reconstruction, which in this case would mean worse preservation of identity. We refer to this in our paper, referencing beta-VAE (Section 4.3, page 8):
>
> "It is challenging to learn a representation that both preserves identity and allows factorisation \cite{higgins2016beta}"
>
> We have indeed tried higher values of alpha and observed how these affect classification and reconstruction. For alpha=1.0, in our model that uses res-nets, the MSE rises to 0.041 (very poor reconstruction) and we generally see no improvement in classification. We point the reviewer to Section 4.3 where we discuss this:
>
> "We found that the naive cVAE-GAN (Bao et al. \cite{bao2017cvae}) failed to synthesise samples with the desired target attribute ‘Not Smiling’. This failure demonstrates the need for models that can deal with both reconstruction and attribute-editing. Note that we achieve good reconstruction by reducing weightings on the KL and GAN loss terms, using \alpha=0.005 and \delta=0.005 respectively."
>
> In most VAE based models there is a trade off between reconstruction and factorization. In our model, factorization comes from the auxiliary loss so the KL term may be weighted less strongly, hence we are able to use a small alpha weighting on the KL term.

---

> ### Author Response · Authors · 2018-11-09
> **Improvements to related work and a request for clarification.**
>
> [Reviewer]
> Note that for example that in beta-VAE it is a similar idea where but it is on z and z|x and the distance used is KL (since it is has closed form with gaussian), min_phi Loss+ beta KL (p(z), p(z|x)), a discussion of the previous related work in the paper is necessary.
>
> [Authors]
> The authors whole heartedly appreciate the contributions beta-VAE has made to the field and specifically to representation learning. As per the reviewer's helpful suggestion, we have added the following to improve the related work section of our paper (adding an additional citation to both beta-VAE and Burgess et al.):
>
> "The beta-VAE \cite{higgins2016beta} objective is similar to the information bottle neck \cite{burgess2018understanding}, minimizing mutual information, I(x;z), which forces the model to exploit regularities in the data and learn a disentangled representation. In our approach we perform a more direct, supervised, factorisation of the latent space, using a mini-max objective, which has the effect of approximately minimizing I(z;y)."
>
> [Reviewer]
> The work is also related to MINE https://arxiv.org/pdf/1801.04062.pdf where one would like to minimize the mutual information I(z;y) this mutual information is estimated through a min/max game.
>
> [Authors]
> Thank you very much for the pointers to additional literature.
>
> We have one question about the connection to MINE (Belghazi et al): To the best of our knowledge, we cannot find the implied connection of MINE with performing an explicit mini-max game, which our paper proposes?
>
> The method presented in the paper, Belghazi et al. 2018, learns a model, T_\theta, that takes set of two variables (e.g. a \in A, b \in B) as input and predicts the mutual information (e.g I(A;B)). According to Algorithm 1 of Belghazi et al., T_\theta is learned via gradient ascent only, there is no mini-max objective for estimating T_\theta. Depending on the application, the mutual information may be minimized or maximized.
>
> Two applications that Belghazi et al. propose include:
> (1) Using T_\theta as a regularizer in a GAN, maximizing the mutual information, I(x;z), between data, x and latent code, z.
>    (a) The only minimax game here is between the generator and discriminator.
>    (b) There is no mini-max objective for estimating T_\theta.
>    (c) The purpose of using T_\theta as a regularizer is to prevent mode dropping.
> (2) T_\theta is used to approximate the mutual information term, I(x;z), in the information bottleneck. In this example, I(x,z), is minimized.
>    (a) There is no minimax game here.
>    (b) There is no mini-max objective for estimating T_\theta.
>
> The only mention of mini-max we found is in Equation 16, which only corresponds to a GAN setting but does not correspond to their method for approximating mutual information.
>
> Additionally, we could not find any example of mutual information computed between a latent code z and a label y, which is our setting.
>
> Could you let us know what you had in mind when making that connection?
>
> Thank you in advance.

---

> > ### Author Response · Authors · 2018-11-22
> > **Improvements to related work (MINE)**
> >
> > We contacted one of the authors of Belghazi et al., who confirmed that they did * not * use a min-max objective to estimate the mutual information. Belghazi et al. approximated mutual information, T, by `maximizing a dual lower bound'. There is a min-max objective if and only if an additional network is incorporated to minimize the mutual information.
> >
> > We are minimizing mutual information without approximating it directly. Our approach is closer to Predictability Minimization (Schmidhuber et al.), where our auxiliary network approximates p(y|z) and the encoder minimizes E_y p(y|z). Note that our auxiliary network takes only z as input, if we were trying to predict mutual information, I(Y;Z), using MINE, the model, T, would take batches of y \in Y and z \in Z as input. In our work we do not approximate mutual information.
> >
> > However, we do agree that it would be suitable to include a reference to Belghazi et al.'s work and have added the following to the related work section:
> >
> > "\cite{belghazi2018MINEMI} proposed a general approach for predicting the mutual information, which may then be minimized via an additional model. Rather than predicting mutual information \citep{belghazi2018MINEMI} between latent representations and labels, we implicitly minimize it via adversarial information factorization."

---

> ### Author Response · Authors · 2018-11-09
> **Improvements to related work and additional comparison with DIP-VAE**
>
> [Reviewer]
> There is a large body of work on disentanglement that the paper does not cite or compare to for instance, InfoGAN,  Beta-VAE https://openreview.net/pdf?id=Sy2fzU9gl and disentangled latent concepts https://arxiv.org/pdf/1711.00848.pdf ([Authors] (DIP-VAE)).
>
> [Authors]
> We appreciate the additional references that the reviewer proposed, two of which we had already included (beta-VAE and InfoGAN). Based on these suggestions we have improved the related work section of our paper by adding the following:
>
> "Finally, while we use labelled data to learn representations, we acknowledge that there an many other models that learn factored, or disentangled, representations from unlabelled data including several VAE variants \citep{higgins2016beta, kumar2018variational}."
>
> We would also like to draw the reviewer's attention to the six instances where we had already cited InfoGAN in our paper and the one instance of beta-VAE.
>
> Below is one quoted example where we investigated the inclusion of a component of our model that was inspired by a similar approach in InfoGAN (Section 4.1, page 6):
>
> "Using \hat{\mathcal{L}}_{class} does not provide any clear benefit. We explored the effect of including this term since a similar approach had been proposed in the GAN literature \cite{chen2016infogan,odena2016conditional} for conditional image synthesis (rather than attribute editing). To the best of our knowledge, this approach has not been used in the VAE literature. This term is intended to maximise I(x,y) by providing a gradient containing label information to the decoder, however, it does not contribute to the factorization of attribute information, y, from \hat{z}."
>
> Though we have cited beta-VAE (Section 4.3, page 8), we did not make a direct comparison to beta-VAE since it is trained without labelled data and a classifier is trained post-hoc; this is similar to the DIP-VAE (an improvement on the beta-VAE). Instead, we chose to compare our results to a state of the art classification model, since it outperforms all other methods, including those whose objective is to learn disentangled representations. For the sake of completeness, below is the comparison of our classification results (which are competitive with state of the art) and those reported in the DIP-VAE paper:
>
> 	Label		|	DIP-VAE		|	ours	|
> -———————————————————————
> Black hair		|	80.6 		|	89.8 	|
> Blonde hair		|	91.9  	        | 	97.2 	|
> Heavy Makeup	|	81.5 		|	92.8 	|
> Wavy hair		|	71.5. 		| 	84.5 	|
> Lipstick. 		|	84.7 		|	94.4 	|
>
> As is clear, our model strongly out-performs DIP-VAE, which is why we did not explicitly compare with these results in our paper, and rather chose to compare to a state of the art classification model. We felt this was a vastly more challenging benchmark.
>
> ---- EDIT ----
>
> We have included additional results comparing our model with DIP-VAE (Kumar et al.) in the Appendix.

---

> ### Author Response · Authors · 2018-11-09
> **Thank you for helping us to improve our paper.**
>
> We would like to sincerely thank the reviewer for reading and understanding our paper and for providing constructive feedback. We have addressed all of the comments below and in one case we respectfully ask for some clarification, please. The feedback from the reviewer has been very helpful for improving our paper (please see the updated version).

---

### Official Review · AnonReviewer1 · 2018-11-01
**Method clarity can be improved, and lacks some key comparisons experimentally.**

**Rating:** 6
**Confidence:** 4

**Review:**

In this paper, the authors introduce a neural network architecture that has three components.
First a VAE is used to encode images in to two latent states \hat{y} and \hat{z}, with \hat{z}
intended to be class (e.g. face attribute) agnostic. The decoder reconstructs images from \hat{y}
and \hat{z} concatenated together. A GAN style discriminator attempts to distinguish the
decoded image from the original input image as real or fake, allowing the decoder to produce
higher quality decoded images. An auxiliary network A attempts to classify the face attribute y
from the class agnostic features \hat{z}, with the idea being that the encoder should try to produce
\hat{z} vectors from which the class cannot be predicted. An additional classifier is trained
using a classification loss \hat{L}_{class} on the encoded reconstructed image, the use of which
I don't understand.

I think additional work on section 2.5 through section 3 would be helpful to improve clarity.
As one example, "y" is unnecessarily overloaded: y denotes a specific attribute, \hat{y}
denotes a latent vector that is intended to not be class agnostic, \tilde{y} denotes the
prediction of an auxiliary network on an intended class-agnostic latent vector \hat{z} of
the presence of the original attribute y, and \hat{\hat{y}} denotes the non agnostic latent
vector achieved by passing the decoded image back through the encoder.

This notational complexity is compounded by the fact that a number of steps in the method are
not well motivated in the text, and left to the reader to understand their purpose. For example,
the authors state that "we incorporate a classification model into the encoder so that our model may
easily be used to perform classification tasks." What does this mean? In the diagram (Figure 1),
where is this classification model? Why in the GAN loss is there a term that compares the
fake loss with the result of classifying a decoded z vector? Is this z \hat{z}, or a latent vector
drawn from a distribution p(z)? If it is the former, how does this term differ from the second
term in the GAN loss. If it is the latter, then shouldn't it be concatenated with some y in order to
be used as input to the decoder D_{\theta}?

Why is it important to extract \hat{\hat{y}} from \hat{x}? In the paper you state that the loss
"provides a gradient containing label information to the decoder," but why can't we use the known label y
of the original input x to ensure that the encoder and decoder preserve this information if it is used as \hat{y}?
Later in the paper, you explicitly state that \hat{\mathcal{L}_{class}} "does not provide any clear benefit."
If that is the case, then you should ideally include it neither in the model nor in the paper. If it was
included primarily because previous models included it, then I would recommend you introduce its use
in a background section on Bao et al., 2017 rather than including it in your model description with an
explanation like "so that our model may easily be used to perform classification tasks."

Ultimately, this last point brings us to a good summary of my concerns with the model: the inclusion
of too many moving parts, some of which the authors explicitly say later on provide no benefit.

Moving on to experimental results, I think this is another area where I have a few concerns. First, in
Figure 2, the authors argue that your model is "better for 6 out of 10 attributes" and comparable results for most others. The authors include a gap of 0.1 in the "Gray_hair" category as "better" but label a gap of 0.5
in the Black hair category as "comparable." I think results in several of the categories are sufficiently close
that error bars would be necessary to draw actual conclusions. If "better" were to mean "better by 0.5" for example,
then the authors method is better on 4 tasks (smiling, blonde hair, heavy makeup, mustache) and worse on 3 (black hair, brown hair, wavy hair).

With respect to the actual attribute editing, my main concern here is a lack of comparison to models other than Bao et al., despite the fact that face attribute changing is an exhaustively studied task. A number of papers like Perarnau et al., 2016, Upchurch et al., 2017, Lample et al., 2017 and others study this task from machine learning perspectives, and in some cases can perform photorealistic image attribute editing without complicated machinery on megapixel face
images. At least the images in Figure 3 and 4 are substantially downsampled from the typical resolution found in the Celeba dataset, suggesting that there was some failure mode on full resolution images.

----

Edit: I've reviewed the authors' addressing my concerns in their paper and am happy to increase my rating as a result.

---

> ### Author Response · Authors · 2018-11-13
> **Our model is no more complex that related models and we compare to a state of art classification model.**
>
> The reviewer's main concerns appear to be complexity of the model and comparison to related work.
>
> [1] Complexity:
> The reviewer's main concern is that the model is too complex, however, our proposed model is no more complex than the accepted paper of Bao et al. Our cost has the same number of components and hyper parameters and our model has the same number of networks (our encoder network has two outputs). Most of our components are also less complex because losses are computed on network outputs rather than on features extracted from multiple intermediate layers. Additionally, we demonstrate that terms in our loss function, \hat{L}_{class}, may be excluded, making our model less complex.
>
> Throughout our work we have been intentionally explicit and detailed about the costs we use. This may have resulted in the complexity of our approach being excessively emphasised, however, it is merely a thorough presentation of our idea, intended to make the work reproducible. Complexity appears to be the reviewer’s main concern, however, since our paper is no more complex than papers previously accepted, we assert that our paper, detailing a novel approach, should be accepted.
>
>
> [2] Comparison to related work:
> The focus of our work has been to learn a representation that factors attribute information from the rest of the representation. We test this factorization process in two ways: (1) attribute editing and (2) attribute classification.
>
> The papers recommended by the reviewer focus only on attribute editing and not on representation learning and they (Upchurch et al., 2017 and Lample et al., 2017) may not be used for, or (Perarnau et al., 2016) have not been demonstrated for attribute classification. To the best of our knowledge, our work is the only approach to learn disentangled representations that may be applied to both image attribute manipulation and simultaneously achieves competitive results with state of the art models on image classification. This novel versatility of the model is certainly a strength of our paper and grounds for acceptance.
>
> When comparing our model to previous work, we chose the most challenging benchmark for facial attribute classification, not just comparing to models intended for attribute editing. Our classification results are highly competitive with this benchmark.
>
> We hope that following the revisions suggested by the reviewer and the inclusion of recommended citations, the reviewer will take our response into consideration and revise their assessment of our paper. Thank you.

---

> ### Author Response · Authors · 2018-11-13
> **Addressing concerns about experimental results.**
>
> [Reviewer]
> Moving on to experimental results, I think this is another area where I have a few concerns. First, in Figure 2, the authors argue that your model is "better for 6 out of 10 attributes" and comparable results for most others. The authors include a gap of 0.1 in the "Gray_hair" category as "better" but label a gap of 0.5 in the Black hair category as "comparable." I think results in several of the categories are sufficiently close hat error bars would be necessary to draw actual conclusions. If "better" were to mean "better by 0.5" for example, then the authors method is better on 4 tasks (smiling, blonde hair, heavy makeup, mustache) and worse on 3 (black hair, brown hair, wavy hair).
>
> [Authors]
> The quotation "better for 6 out of 10 attributes" is no where to be found in our paper. The paper read "outperformed for 6 out of 10". However, we understand the reviewer's concerns and agree that the values are close. For this reason, throughout the paper we have stressed that the results are competitive. We have amended the text in the results section to reflect this also:
>
> "Results in Figure \ref{fig:state_of_art} show that our model is highly competitive with a state of the art facial attribute classifier \cite{zhuang2018multi}. We outperformed by more than 1% on $2$ out of $10$ categories, underperformed by more than 1% on only $1$ category and remained competitive with all other attributes."
>
> [Reviewer]
> With respect to the actual attribute editing, my main concern here is a lack of comparison to models other than Bao et al., despite the fact that face attribute changing is an exhaustively studied task. A number of papers like Perarnau et al., 2016, Upchurch et al., 2017, Lample et al., 2017 and others study this task from machine learning perspectives, and in some cases can perform photorealistic image attribute editing without complicated machinery on megapixel face images.
>
> [Authors]
> We would like to thank the reviewer for pointing out additional related work, this has helped us to improve the related work section of our paper.
>
> The focus of our work has been to learn a representation that factors attribute information from the rest of the representation. We test this factorization process in two ways: (1) attribute editing and (2) attribute classification.
>
> Perarnau et al., 2016 is similar to our model without the L_{aux}, \hat{L}_{class} or \hat{L}_KL. While Perarnau et al. does perform image attribute editing, they do not present classification results.
>
> The method proposed by Upchurch et al., 2017 requires a reverse mapping procedure which is very computationally intensive, applying gradient descent in image space. Additionally, the focus of the work by Upchurch et al., 2017 is not on representation learning and may not be used for image classification.
>
> In our paper, the goal is to learn representations for images. For this goal, our encoder network needs to encode more than just the attribute-invariant information, but also the attribute information itself. The encoder of the Fader Network proposed by Lample et al., 2017 predicts only the attribute-invariant information, while our encoder network predicts both attribute-invariant information and the attribute. This makes our model not only suitable for attribute editing, but it also makes our model suitable for classification. Ultimately, developping models capabable of more than just one task are exciting and important steps forward for the field.
>
> We have updated our paper to include references to Perarnau et al., 2016, Upchurch et al., 2017 and Lample et al., 2017, as per the reviewer's suggestions, to strengthen the related work section of our paper.
>
> [Reviewer]
> At least the images in Figure 3 and 4 are substantially downsampled from the typical resolution found in the Celeba dataset, suggesting that there was some failure mode on full resolution images.
>
> [Authors]
> We use the standard image size, 64x64, and these were used directly without down-sampling. Unfortunately, some resolution was unintentionally lost in Figure 4, when annotated with \hat{y}=0 and \hat{y}=1. We have rectified this and have updated the image with a higher resolution version. However, images in Figure 3 were not affected and we are not aware of any failure modes.
>
> We use images of size 64x64, rather than larger image sizes, (a) to make our ablation study more computationally feasible and (b) to make our results more reproducible by those with modest resources. We believe that this is generally good for the field.
>
> We provide extensive quantitative results which show (a) that the attribute manipulation is reliable and (b) that we achieve a low reconstruction error. Additionally, our classification results are further evidence that our model does indeed factor attribute information from the rest of the latent vector, which is our objective.

---

> > ### Comment · AnonReviewer1 · 2018-11-13
> > **More responses**
> >
> > > The quotation "better for 6 out of 10 attributes" is no where to be found in our paper. The paper read "outperformed for 6 out of 10".
> >
> > Okay.
> >
> > > However, we understand the reviewer's concerns and agree that the values are close...
> >
> > Thanks. I do feel that the updated text more honestly and scientifically represents the results of Figure 2.
> >
> > > The focus of our work has been to learn a representation that factors attribute information from the rest of the representation. We test this factorization process in two ways: (1) attribute editing and (2) attribute classification.
> >
> > I agree that you present results on both attribute classification and attribute editing. My concern is whether it is clear that you perform either task significantly better than state of the art methods in either task. I focused on attribute editing papers simply because this is a case where I think there are very strong baselines that work on megapixel images.
> >
> > You argue that you use smaller images "(a) to make our ablation study more computationally feasible and (b) to make our results more reproducible by those with modest resources." I definitely agree it is useful to include the smaller results for this purpose. However, my concern is that a lack of any results on larger face images makes it difficult to compare your approach with methods that succeed at editing larger images.

---

> > > ### Author Response · Authors · 2018-11-14
> > > **We have included comparison to IcGAN**
> > >
> > > [Reviewer]
> > >
> > > You argue that you use smaller images "(a) to make our ablation study more computationally feasible and (b) to make our results more reproducible by those with modest resources." I definitely agree it is useful to include the smaller results for this purpose. However, my concern is that a lack of any results on larger face images makes it difficult to compare your approach with methods that succeed at editing larger images.
> > >
> > > [Authors]
> > >
> > > For work that focuses only on attribute editing, it may make sense to consider higher resolution images, however we focus on representation learning, where it is common to used images at resolution 64x64 (or less) [higgins2016beta, bao2017cvae, li2017alice, larsen2016autoencoding, burgess2018understanding, kumar2018variational]. Of the three papers you propose (Upchurch et al., Lample et al., Perarnau et al.) none of them are motivated by representation learning and only Perarnau et al. proposes an encoder that outputs a representation that is sufficient to describe an input image.
> > >
> > > As per your suggestion, we have included a comparison with IcGAN (Perarnau et al.) in Tables 1 and 3, taking values from Lample et al. We compare to IcGAN since they also train on images that are 64x64. Our model without residual layers obtains that same reconstruction error as IcGAN, 0.028, while our model with residual layers achieves a much lower reconstruction error, 0.011. Lample et al. also suggests that the IcGAN only successfully edits attributes (Smiling --> Not Smiling) 9.9% of the time, while our model (with residual layers) successfully edits them at least 98% of the time. Our model without residual layers edits successfully edits them 81% of the time.
> > >
> > > [Please note that there is a typo in the Lample et al. paper, the authors write RMSE rather than MSE. We have contacted the authors and they confirm that this was a typo and they were in fact reporting MSE.]
> > >
> > > Thank you for helping us to improve our paper with additional comparisons to related work.

---

> > > ### Author Response · Authors · 2018-11-14
> > > **Our results are sufficient to confirm that our model achieves good factorization -- the objective of our paper.**
> > >
> > > [Reviewer]
> > > I agree that you present results on both attribute classification and attribute editing. My concern is whether it is clear that you perform either task significantly better than state of the art methods in either task.
> > >
> > > [Authors]
> > >
> > > Our work presents a method for learning representations that factor the attribute information from the rest of the latent representation. To evaluate our factorisation method we perform facial attribute editing and classification. Our results are (more than) sufficient to confirm that our model achieves good factorization -- it is not necessary for our model to achieve state of the art results to confirm this.
> > >
> > > We do indeed claim that our results are competitive with a state of the art classification, but we do not claim to present state of art classification results. Please note that if we were aiming to achieve state of the art classification we would, for example, have used much deeper networks,  Zhuang et al. 2018 use 13 layers while our encoder (which we use for classification) has only seven.

---

> ### Author Response · Authors · 2018-11-13
> **The encoder acts as a classifier, outputting an attribute vector, hat{y}, as well as a latent vector, z.**
>
> [Reviewer]
> Why is it important to extract \hat{\hat{y}} from \hat{x}? In the paper you state that the loss "provides a gradient containing label information to the decoder," but why can't we use the known label y of the original input x to ensure that the encoder and decoder preserve this information if it is used as \hat{y}?
>
> [Authors]
> L_{class} ensures that \hat{y} contains label information, but this loss is not dependant on the parameters of the decoder and, therefore, cannot be used to update the decoder. Note that there is a precendence for computing \hat{\hat{y}}: the Bao et al. model also use it to provide label information to the decoder. We could have placed an additional classifier at the output of the decoder, as is done by Bao et al., to compute \hat{\hat{y}}. However, rather than introducing and training another classifier, we made use of our encoder which is already able to predict labels, thus we pass the reconstructed image back through the encoder.
>
> [Reviewer]
> Later in the paper, you explicitly state that \hat{\mathcal{L}_{class}} "does not provide any clear benefit." If that is the case, then you should ideally include it neither in the model nor in the paper. If it was included primarily because previous models included it, then I would recommend you introduce its use in a background section on Bao et al., 2017 rather than including it in your model description with an explanation like "so that our model may easily be used to perform classification tasks."
>
> [Authors]
> The explanation for using \hat{L}_{class} is not "so that our model may easily be used to perform classification tasks.", it is because  (Section 3.1) it "provides a gradient containing label information to the decoder".
>
> We chose to investigate the need for \hat{L}_{class} because other works had proposed a similar idea - that is to train a classifier on reconstructed samples \cite{odena2016conditional, bao2017cvae}. We did not know a priori that this component would be redundant, and only discovered this following investigation. Indeed, it is a useful and relevant finding of our work for the representation learning community. Rather than simply leaving this component out of our model, we chose to perform an extensive ablation study to provide evidence for this.
>
> We would like to make a final note concerning the quotation "so that our model may easily be used to perform classification tasks". This was simply referring to the fact that the encoder outputs an attribute label vector, \hat{y}, and hence may be used as a classifier. As mentioned above, we have amended this section of the text (Section 3.1) to read:
>
> "Additionally, the encoder also acts as a classifier, outputting an attribute vector, \hat{y}, along side a latent vector, \hat{z}."
>
> [Reviewer]
> Ultimately, this last point brings us to a good summary of my concerns with the model: the inclusion of too many moving parts, some of which the authors explicitly say later on provide no benefit.
>
> [Authors]
> Since our model is no more complex than that of Bao et al., an accepted paper, we assert that this is not grounds for rejection. The original Bao et al. model consists of 6 components (see Equation 7 of Bao et al.). There is only one term in our loss function that following investigation, we considered to be redundant. Rather than just leaving this out, we performed an extensive ablation study to provide evidence for this.
>
> The source of the confusion seems to be summed up by the question, "where is this classification model?". While the reviewer has understood the core contributions of our paper, this misunderstanding has lead to most of the questions above. Simply, the encoder acts as a classifier because it predicts both an attribute, \hat{y}, and a latent vector, \hat{z}. We have amended our paper to make this more clear, adding the following to Section 3.1:
>
> "Additionally, the encoder also acts as a classifier, outputting an attribute vector, \hat{y}, along side a latent vector, \hat{z}."
>
> Crucially, the encoder in our model may be used as a classifier, unlike in other attribute editing models and we demonstrate that our classifier achieves results that are competitive with state of the art classification results.

---

> > ### Comment · AnonReviewer1 · 2018-11-13
> > **More responses**
> >
> > > Since our model is no more complex than that of Bao et al., an accepted paper, we assert that this is not grounds for rejection.
> > > The explanation for using \hat{L}_{class} ...
> >
> > I covered \hat{L}_{class} in a comment above, but since you are addressing it again, I am happy to as well.
> >
> > First, I disagree with your assertion. In my opinion, presenting a model with moving parts that you claim *in the paper* serve no purpose adds needless complexity.
> >
> > In my opinion, components of your model that do nothing should not be included in the full description of your model. It  strictly adds unnecessary complexity, and the section should be a description of the full model you are proposing, not every component that was tried along the way.
> >
> > The ablation study is great, but it would suffice to mention in a single experimental result that following other literature you tried adding such a term, but as Table 1 demonstrates it didn't help and therefore was excluded from the model.
> >
> > The fundamental problem with including components like this just because other papers did is that useless components of models propagate this way. If you feel your paper is the one to finally discover that it is useless, then great! Your paper should be the first to not include it as part of the model.

---

> > > ### Author Response · Authors · 2018-11-14
> > > **Thank your helping us to simplify our model.**
> > >
> > > We have now revised our paper and we do not include \hat{L}_{class} in our model description. As per your helpful suggestion, we now have add a single experiment in our ablation study to demonstrate that \hat{L}_{class} was not helpful.
> > >
> > > Thank you for this suggestion and for helping us to improve our paper.

---

> ### Author Response · Authors · 2018-11-13
> **Explaining the `classification model'.**
>
> [Reviewer]
> I think additional work on section 2.5 through section 3 would be helpful to improve clarity. As one example, "y" is unnecessarily overloaded: y denotes a specific attribute, \hat{y} denotes a latent vector that is intended to not be class agnostic, \tilde{y} denotes the prediction of an auxiliary network on an intended class-agnostic latent vector \hat{z} of the presence of the original attribute y, and \hat{\hat{y}} denotes the non agnostic latent vector achieved by passing the decoded image back through the encoder.
>
> [Authors]
> We believe our notation is consistent, correct and necessary, and this reviewer (as well as the others) have understood our model. We provide a clear diagram as well as an algorithm and descriptions in the text.
>
> [Reviewer]
> This notational complexity is compounded by the fact that a number of steps in the method are not well motivated in the text, and left to the reader to understand their purpose.
>
> [Authors]
> Our core contribution is the introduction of an auxiliary network for factorizing attribute information from the rest of the latent code. This is communicated clearly in Section 3.2 and the reviewer appears to understand the * motivation * for the auxiliary network well:
>
> Reviewers own words: "An auxiliary network A attempts to classify the face attribute y from the class agnostic features \hat{z}, with the idea being that the encoder should try to produce \hat{z} vectors from which the class cannot be predicted."
>
> We incorporate our network into a pre-existing model, the VAE-GAN which is also communicated clearly in Section 2. In Section 2 we introduce VAEs and GANs, and explain the benefits of combining the two in Section 2.3 'Best Of Both GAN And VAE'. Finally, we also explain conditional GANs. The VAE, GAN and VAE-GAN are previous related works, motivated by their respective authors.
>
>
> [Reviewer]
> For example, the authors state that "we incorporate a classification model into the encoder so that our model may easily be used to perform classification tasks." What does this mean? In the diagram (Figure 1), where is this classification model?
>
> [Authors]
> We would like the thank the reviewer for raising this concern. To avoid confusion, we have amended the paper in Section 3.1 to read as follows:
> "Additionally, the encoder also acts as a classifier, outputting an attribute vector, \hat{y}, along side a latent vector, \hat{z}."
>
> There is no separate classification network. The encoder, which takes an image, x, as input, is split in two, outputting a latent vector \hat{z} and an attribute vector \hat{y}. The model is trained such that the attribute vector may be used to classify an input image, x, therefore the encoder is also a classifier.
>
> This is described in Section 2.4 (quote from our paper):
> “the encoder outputs both a latent vector, \hat{z}, and an attribute vector, \hat{y}”
>
> Our point here, is that rather than adding a separate classifier that takes reconstructed images as input, as is done in Bao et al. (illustrated in our paper in Figure 1(b)), we incorporate the classifier into the encoder, E_{y, \phi}, (illustrated in Figure 1(a)) which allows our model to be used both for image attribute editing and classification. This also avoids the need to train an entirely separate classifier network. We are essentially killing two birds with one stone in this proposed approach.
>
> [Reviewer]
> Why in the GAN loss is there a term that compares the fake loss with the result of classifying a decoded z vector?
>
> [Authors]
> There are three terms in the GAN loss. In the first, the discriminator, C_\chi, is applied to real samples. In the second and third term the discriminator is applied to fake images. There are two sources of fake images; (1) the reconstructed images, D_\theta(E_\phi(x)), and (2) synthetic images, D_\theta(E_{\phi,z}(x), y). The GAN loss is similar to the one used by Bao et al. (see Line 9 of Algorithm 1 of Bao et al.).
>
> [Reviewer]
> Is this z \hat{z}, or a latent vector drawn from a distribution p(z)?
>
> [Authors]
> By definition z is drawn from the prior, p(z), this is described in Section 2.3 when describing VAE-GAN; “latent variable z, which is drawn from a specified random distribution, p(z)”. This is also illustrated in line 5 of Algorithm 1. We have made this clearer by adding the following after defining L_{gan}: “and z \sim p(z)”.
>
> [Reviewer]
> If it is the former, how does this term differ from the second term in the GAN loss. If it is the latter, then shouldn't it be concatenated with some y in order to be used as input to the decoder D_{\theta}?
>
> [Authors]
> We would like to thank the reviewer for catching this typo. Indeed, there is supposed to be a y input to the decoder, we have updated the paper to reflect this, the third term now reads: “L_{bce}(y_{fake}, C_\chi(D_\theta(z, y)))]”.

---

> ### Author Response · Authors · 2018-11-13
> **"There is no additional classifier in our model. The encoder itself acts as a classifier"**
>
> [Authors]
> We would like to sincerely thank the reviewer for reading our paper and for providing constructive feedback. The reviewer appears to have understood all the components of our model well, with the exception of the \hat{L}_{class} loss, which has been the focus of majority of the comments. We have addressed all of the reviewer's comments below, as well as improved our paper where necessary (please see the updated version).
>
> [Reviewer]
> An additional classifier is trained using a classification loss \hat{L}_{class} on the encoded reconstructed image, the use of which I don't understand.
>
> [Authors]
> There is no additional classifier in our model. The encoder itself acts as a classifier, predicting both a latent vector \hat{z}, and a label \hat{y}. The primary loss for ensuring that the encoder is an effective classifier is L_{class}, not \hat{L}_{class} as claimed by the reviewer.
>
> With regards to \hat{L}_{class}, the following quote from our paper (Section 3.1) explains its purpose:
> “The classification loss, \hat{L}_{class}, provides a gradient containing label information to the decoder, which otherwise the decoder would not have \citep{chen2016infogan}.”
>
> We found that this term did not play an important role in our model. This is stated in our paper as follows (Section 4.2):
> “\hat{L}_{class} does not provide any clear benefit. We explored the effect of including this term since a similar approach had been proposed in the GAN literature \cite{chen2016infogan,odena2016conditional} for conditional image synthesis.”

---

> > ### Comment · AnonReviewer1 · 2018-11-13
> > **My point is that \hat{L}_{class} should not be in the paper at all.**
> >
> > With regards to the purpose of \hat{L}_{class}, the quote you provide is clearly not sufficient to adequately explain its purpose. Evidence for this is the fact that, as you later demonstrate with an ablation study, the loss has no purpose.
> >
> > In my opinion, when explaining your model, you should not include terms that "did not play an important role in [your] model" and "[do] not provide any clear benefit.", *regardless of whether those terms have been used in prior art.* If you feel mentioning this term is sufficient, a simple explanation of why it *wasn't* included would suffice.

---

> > > ### Author Response · Authors · 2018-11-14
> > > **We have removed \hat{L}_{class} from our model description -- simplifying our model.**
> > >
> > > We have now revised our paper and we do not include \hat{L}_{class} in our model description. As per your helpful suggestion (below), we have added a single experiment in our ablation study to demonstrate that \hat{L}_{class} was not helpful.

---

> > > > ### Comment · AnonReviewer1 · 2018-11-14
> > > > **Thank you for addressing my concerns, I have increased my rating.**
> > > >
> > > > Hi authors,
> > > >
> > > > Thank you for addressing my concerns in the text of the paper.
> > > >
> > > > I do feel that the paper is overall much improved, particularly the model explanation with one fewer moving component. I have increased my score to be in line with the other reviewers.

---

### Public Comment · (anonymous) · 2018-10-12
**Why not cite FaderNetworks in NIPS2017**

I just wonder why not cite "Fader Networks:Manipulating Images by Sliding Attributes" in NIPS2017.  In my opinion,  The adversarial factorisation in latent space in this paper is quite similar with the referred paper.  Both aim to disentangle the attribute-invariant representations and the attribute labels via an adversarial classifier.

---

> ### Author Response · Authors · 2018-10-12
> **We predict both the *attribute* and the attribute-invariant information, making our model suitable for classification.**
>
> Hello,
>
> Thank you for showing interest in our paper and for pointing us to this work. We agree that there are some similarities between our work, however there is also a key difference.
>
> In our paper, the goal is to learn representations for images, for this our encoder network needs to encode more than just the attribute-invariant information, but also the attribute information itself. The encoder of the Fader Network predicts only the attribute-invariant information, while our encoder network predicts both attribute-invariant information and the attribute. This makes our model not only suitable for synthesising images, but it also makes our model suitable for classification.
>
> We present classification results, using our model, that are highly competitive with state of the art classification results of  Zhuang et al. (2108).
>
> (Additional details)
>
> As mentioned above, unlike in the Fader Networks, our encoder model predicts attribute information, \hat{y}, along side the attribute-invariant information, \hat{z}, which we refer to as identity. During training, our decoder network is fed with predicted attribute values, \hat{y}, rather than the ground truth values, y, as in the Fader Networks. Training the decoder to reconstruct images from predicted attributes, \hat{y}, combined with the adversarial factorization, forces the encoder network to put attribute information into \hat{y}, resulting in the encoder being an excellent classifier.
>
> We will add Fader Networks to the related work section of our paper in the next revision.

---

### Meta-Review · Area_Chair1 · 2018-12-15
**Good results on classification and attribute manipulation but has considerable overlap with Fader networks**

**Confidence:** 5
**Recommendation:** Reject

**Metareview:**

The paper proposes a supervised adversarial method for disentangling the latent space of a VAE into two groups: latents z which are independent of the given attribute y, and \hat{y} which contains information about y. Since the encoder also predicts \hat{y} it can be used for classification and the paper shows competitive results on this task, apart from the attribute manipulation task. Reviewers had raised points about model complexity and connections to prior works which the authors have addressed and the paper is on the borderline based on the scores.

Though none of the reviewers explicitly pointed out the similarity of the paper with Fader networks (Lample et al., 2017), the adversarial setup for getting attribute invariant 'z' is exactly same as in Fader networks, as also pointed out in an anonymous comment. The only difference is that encoder in the current paper also predicts the attribute itself (\hat{y}), which is not the case in Fader n/w, and hence the encoder can be used as a classifier as well (authors have also mentioned and discussed this difference in their response). However, the core idea of the paper as outlined in the title of the paper, ie, using adversarial loss for information factorization, is very similar to this earlier work, which diminishes the originality of the work.

With the borderline review scores, the paper can go in either of the half-spaces (accept/reject) but I am hesitant to recommend an "accept" due to limited originality of the approach. However, if there is space in the program, the paper can be accepted.